# An amide to thioamide substitution improves the permeability and bioavailability of macrocyclic peptides

Pritha Ghosh [1], Nishant Raj [1], Hitesh Verma[1], Monika Patel[2,3], Sohini Chakraborti[1], Bhavesh Khatri[1], Chandrashekar M. Doreswamy[4], S. R. Anandakumar[4], Srinivas Seekallu[4], M. B. Dinesh[5], Gajanan Jadhav[6], Prem Narayan Yadav[2,3] & Jayanta Chatterjee [1]✉

Solvent shielding of the amide hydrogen bond donor (NH groups) through chemical modification or conformational control has been successfully utilized to impart membrane permeability to macrocyclic peptides. We demonstrate that passive membrane permeability can also be conferred by masking the amide hydrogen bond acceptor ($>C = O$) through a thioamide substitution ($>C = S$). The membrane permeability is a consequence of the lower desolvation penalty of the macrocycle resulting from a concerted effect of conformational restriction, local desolvation of the thioamide bond, and solvent shielding of the amide NH groups. The enhanced permeability and metabolic stability on thioamidation improve the bioavailability of a macrocyclic peptide composed of hydrophobic amino acids when administered through the oral route in rats. Thioamidation of a bioactive macrocyclic peptide composed of polar amino acids results in analogs with longer duration of action in rats when delivered subcutaneously. These results highlight the potential of O to S substitution as a stable backbone modification in improving the pharmacological properties of peptide macrocycles.

The amide bond is one of the most prevalent functional groups in pharmaceuticals and bioactive compounds with unparallel utility in covalently linking multiple pharmacophore fragments and engaging in high-affinity target binding through hydrogen bonds[1]. At the same time, the amide bond in peptides composed of canonical amino acids is susceptible to proteolysis by peptidases. The proteolytic susceptibility, coupled with the low permeability of amide bonds across the biological membrane, is a major deterrent in peptide drug discovery[2]. Thus, macrocyclization has emerged as an attractive alternative to impart drug-like properties to peptides[3]. The unique structural features of macrocyclic peptides arising from restricted conformational freedom and local secondary structure motifs allow them to adopt bioactive conformations with remarkable potency and selectivity[4–6]. Tremendous progress in synthetic[7] and display techniques over the last decade has simplified the discovery of functional macrocyclic peptides against any target[8]. However, strategies to improve the pharmacological properties like metabolic stability, membrane permeability, and oral bioavailability of generic peptides have been slow to emerge[9].

Reduction in the number of H-bond donors (-NHs) to facilitate the passive permeability of peptides through conformational control[10,11], steric occlusion[12], and N-methylation[13–15] of the amide bond have been the commonly sought-after methods. To date, N-methylation is the most dominant chemical modification to improve the

[1]Molecular Biophysics Unit, Indian Institute of Science, Bangalore 560012 Karnataka, India. [2]Neuroscience & Ageing Biology, CSIR-CDRI, Lucknow 226031 Uttar Pradesh, India. [3]Academy of Scientific and Innovative Research (AcSIR), Ghaziabad 201002, India. [4]Department of Pre-clinical Research, Anthem Biosciences Pvt. Ltd., Bangalore 560099 Karnataka, India. [5]Central Animal Facility, Indian Institute of Science, Bangalore 560012 Karnataka, India. [6]Eurofins Advinus Biopharma Services India Pvt. Ltd., Bangalore 560058 Karnataka, India. ✉e-mail: jayanta@iisc.ac.in

pharmacological properties of macrocyclic peptides[16–21]. The success of N-methylation owes much to the favorable pharmacological properties of naturally occurring N-methylated macrocyclic peptide Cyclosporin A[22,23]. From a physicochemical standpoint, N-methylation facilitates the passive permeability of peptides across the intestinal membrane by lowering the desolvation penalty of amide bonds and providing protease resistance by masking the amide protons[24]. Nonetheless, N-methylation introduces conformational flexibility in the peptide backbone by a) removing the amide protons (HNs) that engage in intramolecular H-bonds and b) lowering the $(O)C-N(CH_3)$ rotational barrier, which negatively impacts the bioactivity of macrocyclic peptides[25]. Thus, alternative strategies to improve the pharmacological properties of peptide macrocycles while retaining their bioactive conformation are much needed.

A comprehensive analysis of hydrogen bonding in protein crystal structures by Baker and Hubbard reveals that a significantly large number of >C=O groups than NH groups make two or more hydrogen bonds[26]. This emanates from the ability of carbonyl oxygen in amide bonds to utilize its two lone pairs as H-bond acceptors over a wide angular range. In contrast, amide nitrogen acts as a single hydrogen donor, where the H-bond acceptor is colinear with the N-H bond. This suggests that masking the H-bond acceptor, >C=O, in amide bonds should significantly reduce its interaction with solvent water molecules and impart lipophilicity to peptides. We recently found that the S atoms of >C=S are less solvated and weakly bound to the binding site water molecules than the O atoms of >C=O[27]. Therefore, we sought to substitute a single H-bond acceptor, >C=O, in macrocyclic peptides with the isostere >C=S. Herein we show that the substitution of a single amide (-CONH-) with a thioamide (-CSNH-) in a macrocyclic peptide composed of nonpolar amino acid side chains, significantly improves the passive transcellular permeability and metabolic stability, consequently enhancing its plasma exposure post oral administration in rats. We also demonstrate the potential of this single-atom substitution to derive analogs of a bioactive macrocyclic peptide composed of polar amino acid side chains that show a significantly longer duration of action in vivo. Our study demonstrates that masking the amide hydrogen bond acceptors with >C=S is a viable and alternative strategy to N-methylation in improving the pharmacological properties of macrocyclic peptides.

## Results

### HBA-masking improves the lipophilicity of peptides

The H-bond acceptor (HBA), >C=O, unlike the H-bond donor (HBD), NH, in amide bonds has never been targeted to modulate the permeability of peptides. Thus, we adopted a bottom-up approach by initially estimating the lipophilicity of an isolated amide bond on N-methylation (HBD-masking) and thioamidation (HBA-masking). A preliminary assessment of lipophilicity derived by calculating the AlogP values, which is determined from a regression model based on the atomic lipophilicity of compounds[28], indicate that the thioamidated dipeptides (1a–7a) are more lipophilic than their respective amide (1–7) and N-methylated (1b–7b) analogs (Fig. 1a and Supplementary Fig. 1a). We next experimentally determined their retention in a hydrophobic column (C18) through HPLC (Supplementary Fig. 1b) and water-octanol distribution coefficient ($logD_{7.4}$) (Fig. 1a). Irrespective of the polarity of dipeptides 1–7, we noted a significant increase in the C18 retention time and $logD_{7.4}$ of 1a–7a and 1b–7b indicating their enhanced lipophilicity. This suggests that HBA and HBD masking in this model dipeptide have a comparable effect on lipophilicity. The amide bonds in macrocyclic peptides engage in intra- and intermolecular H-bonds; thus, we wondered if thioamidation of >C=O irrespective of its involvement in intra/intermolecular H-bond would increase the lipophilicity of macrocyclic peptides. Towards this goal, we synthesized the regioisomers of monothionated cyclo(-D-Ala-L-Ala_4-) (8)[29] and cyclo(-D-Ala-L-Ala_5-) (9) and determined their

lipophilicity. We observe that in both the polar cyclocpenta- and cyclohexapeptide scaffolds, a single O to S substitution significantly increase the lipophilicity of the parent macrocycle (Fig. 1b and Supplementary Fig. 2).

### Impact of thioamidation on passive permeability of macrocyclic peptides

Since lipophilicity shows a strong positive correlation with cellular permeability of non-peptidic macrocycles[30], we sought to assess if this single atom substitution could improve the passive transcellular permeability of macrocyclic peptides. To test this, we chose three macrocyclic peptides cyclo(-D-Leu$^1$-Leu-Leu-D-Pro-Tyr-Leu$^6$-)[31] (6-mer) (10), cyclo(-Ile$^1$-Ala-Ala-Phe-Pro-Ile-Pro$^7$-)[12] (7-mer) (11), and cyclo(-D-Leu$^1$-Leu-D-Pro-D-Leu-Leu-D-Ala-Pro-Leu$^8$-)[10] (8-mer) (12) with varying size, composition, and lipophilicity (AlogP) (Fig. 1c). We synthesized their monothionated regioisomers 10a–10f, 11a–11g, 12a–12h (Supplementary Table 1) and assessed the aqueous solubility, retention in the C18 column, and $logD_{7.4}$ to evaluate the impact of thioamidation (Supplementary Fig. 3b, c and Fig. 1d). In all the three scaffolds, thioamidation led to reduced aqueous solubility and enhanced retention in the C18 column. Furthermore, these two parameters correlated well (Supplementary Fig. 3d–e) with the octanol-water distribution coefficient ($logD_{7.4}$) indicating enhanced lipophilicity of the macrocyclic scaffolds on thioamidation, irrespective of their size and composition. Encouraged by this, the passive permeability of the analogs was evaluated using an artificial membrane (PAMPA), which measures permeability of compounds in the absence of transporters and efflux systems[32]. In all three scaffolds, thioamidation resulted in a significant increase in the permeability of monothioamidated analogs (Fig. 1e). Although the absolute permeabilities of 11a–11g are lower than 10a–10f and 12a–12h, the relative change in permeability of 11a–11g with respect to 11, is comparable to the other two scaffolds (Supplementary Table 2). This suggests that thioamidation can improve the passive permeability of macrocycles with varying lipophilicity. However, the absolute permeability of the thioamidated macrocycle scales with lipophilicity ($logD_{7.4}$) of the parent macrocycle, which is dictated by its composition and not the ring size (Fig. 1f). Nevertheless, a large increase in lipophilicity (e.g. 12c) can potentially limit the passive permeability of macrocyclic peptides, as observed before[33]. It is also important to note that 12, referred as D8.1 by Bhardwaj et al. displayed the highest permeability within the computationally designed 8-mer scaffolds due to the maximisation of intramolecular H-bond satisfaction[10]. However, the enhanced permeability of 12 through thioamidation (12d–12h) strongly justifies the potential of HBA masking to further navigate the permeability space beyond the currently available toolkit.

We next sought to investigate the origin of permeability on thioamidation of 10, a scaffold that has been extensively investigated. As noted in the previous reports, 10 showed a moderate permeability ($2.45 \times 10^{-6}$ cm/s), which significantly improved on thioamidation at D-Leu1 (10a), Tyr5 (10e), and Leu6 (10f) with marginal improvement at the other sites (10b–10d) (Fig. 1e). What was striking is the marked improvement in permeability of 10a ($10.75 \times 10^{-6}$ cm/s) by this single atom substitution against PC ($15.76 \times 10^{-6}$ cm/s), derived by altering the amino acid chirality at 3 sites[31] (Fig. 1e). As hydrogen bonding and solvation of NH groups play an important role in the permeability of macrocycles, we determined the solvent accessibility of individual amide protons in 10 and 10a–10f by hydrogen-deuterium exchange (HDX) through NMR spectroscopy (Figs. 2a–c). Since the cyclic peptides have moderate solubility in water (Supplementary Fig. 3c), we used DMSO-$d_6$:$D_2O$ (8:2) to mimic the aqueous polar environment and $CDCl_3$:$CD_3OD$ (8:2) as a membrane mimic, where $D_2O$ and $CD_3OD$ act as the deuterium donor, respectively. We note that thioamidation at D-Leu1 (10a), Tyr5 (10e), and Leu6 (10f) results in slow exchange of the amide protons with respect to 10, indicative of enhanced solvent

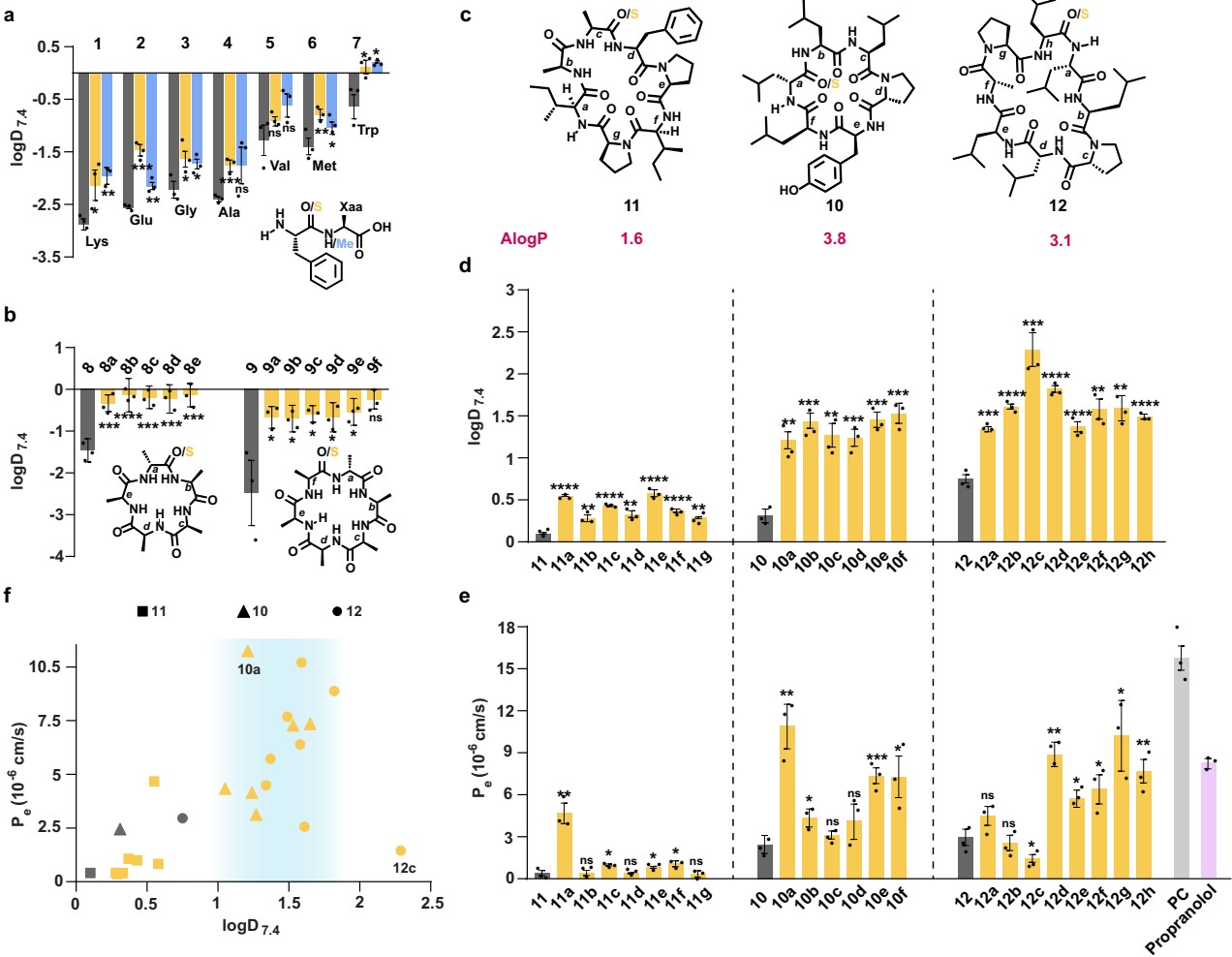

**Fig. 1 | Effect of O to S substitution on the lipophilicity and permeability of peptides. a** Octanol-water partition coefficient (logD$_{7.4}$) of oxo (grey), thioami-dated (yellow), and N-methylated (blue) dipeptides with the common sequence FX. The "X" residues are mentioned at the bottom of the bars. **b** logD$_{7.4}$ of the cyclic pentaalanine (**8**) and hexaalanine (**9**) peptides with their respective thioamidated analogs indicated by small letters, which denote the site of thioamidation. **c** AlogP of cyclo(-D-Leu$^1$-Leu-Leu-D-Pro-Tyr-Leu$^6$-)(**10**), cyclo(-Ile$^1$-Ala-Ala-Phe-Pro-Ile-Pro$^7$-)(**11**), and cyclo(-D-Leu$^1$-Leu-D-Pro-D-Leu-Leu-D-Ala-Pro-Leu$^8$-)(**12**). The small letters denote the site of thioamidation in the individual scaffolds. **d** logD$_{7.4}$ of **10**, **11**, **12** and their respective thioamidated analogs. **e** Membrane permeability of the mac-rocyclic peptides determined by the PAMPA (P$_e$). Propranolol and **PC** (cyclo(-D-Leu$^1$-Leu-D-Leu-Pro-Tyr-Leu$^6$))[31] were used as the markers for transcellular trans-port. **f** The plot of logD$_{7.4}$ vs. PAMPA permeability of all the 24 macrocyclic pep-tides. The blue shaded region highlights the lipophilicity zone of macrocycles with permeability > 2.5 ×10$^{-6}$ cm/s; for clarity only the mean values are plotted. $n = 3 \pm$ SEM. Each bar represents mean values of three biological replicates: dots are individual data points. Statistical significance of the analogs was measured against the (all-amide) parent molecule by a one-tailed unpaired $t$-test. *$p < 0.05$, **$p < 0.01$, ***$p < 0.001$, ****$p < 0.0001$, ns (non-significant) > 0.05. $p$ (**1** vs **1a, 1b**) = 0.0339, 0.0045; $p$ (**2** vs **2a, 2b**) = 0.0003, 0.002; $p$ (**3** vs **3a, 3b**) = 0.0213, 0.0257; $p$ (**4** vs **4a, 4b**) = 0.001, 0.0603; $p$ (**5** vs **5a, 5b**) = 0.1513, 0.0716; $p$ (**6** vs **6a, 6b**) = 0.0084, 0.0418; $p$ (**7** vs **7a, 7b**) = 0.021, 0.0105; $p$ (**8** vs **8a, 8b 8c, 8d, 8e**) = 0.002, <0.0001, 0.0001, 0.0006, 0.0005; $p$ (**9** vs **9a, 9b 9c, 9d, 9e, 9f**) = 0.0308, 0.0310, 0.0267, 0.0308, 0.0237, 0.0154; $p$ (**10** vs **10a, 10b, 10c, 10d, 10e, 10f**) = 0.0011, 0.0003, 0.0014, 0.0009, 0.0003, 0.0005; $p$ (**11** vs **11a, 11b, 11c, 11d, 11e, 11f, 11g**) = <0.0001, 0.0050, <0.0001, 0.0016, <0.0001, <0.0001, 0.0027; $p$ (**12** vs **12a, 12b, 12c, 12d, 12e, 12f, 12g, 12h**) = 0.0002, <0.0001, 0.0007, <0.0001, <0.0001, 0.0022, 0.0014, <0.0001 for logD$_{7.4}$. $p$ (**10** vs **10a, 10b, 10c, 10d, 10e, 10f**) = 0.0019, 0.0239, 0.0503, 0.0815, 0.0005, 0.016; $p$ (**11** vs **11a, 11b, 11c, 11d, 11e, 11f, 11g**) = 0.0019, 0.4776, 0.0114, 0.4596, 0.0299, 0.0207, 0.3486; $p$ (**12** vs **12a, 12b, 12c, 12d, 12e, 12f, 12g, 12h**) = 0.0839, 0.3126, 0.0344, 0.0017, 0.0130, 0.0199, 0.0123 for P$_e$ by PAMPA. Source data are provided as a Source Data file.

shielding. In comparison, a fast exchange indicating solvent exposure of the amide protons is observed for **10b, 10c,** and **10d** in DMSO-$d_6$ (Fig. 2b and Supplementary Fig. 4). In CDCl$_3$, the amide protons exchange at a slower rate than in DMSO-$d_6$, suggesting their solvent-shielded nature. Most amide protons in **10a** are solvent-shielded in the nonpolar environment, followed by **10d, 10e,** and **10f**, while **10b** and **10c** show faster exchange than **10** (Fig. 2c and Supplementary Fig. 5).

These results indicate that **10a, 10e,** and **10f** have a higher ten-dency than **10** to transfer from a polar to a nonpolar environment. Once in a nonpolar environment, the adoption of a membranophilic conformation wherein the amide protons are solvent-shielded

facilitates the translocation of the macrocyclic peptide. On the con-trary, the high solvent exposure of amide protons in **10b, 10c,** and **10d** as compared to **10** in DMSO-$d_6$ suggests a lower tendency of these three analogs to transfer into a nonpolar environment. Moreover, the high solvent exposure of **10b** and **10c** in CDCl$_3$ is suggestive of a polar conformation within a nonpolar environment, which would be dis-favored. Thus, thioamide substitution increases the passive perme-ability of **10a, 10e,** and **10f** by reducing the solvent exposure of amide protons and facilitating their transfer from the polar exterior to the nonpolar interior of the membrane with the adoption of a membra-nophilic conformation.

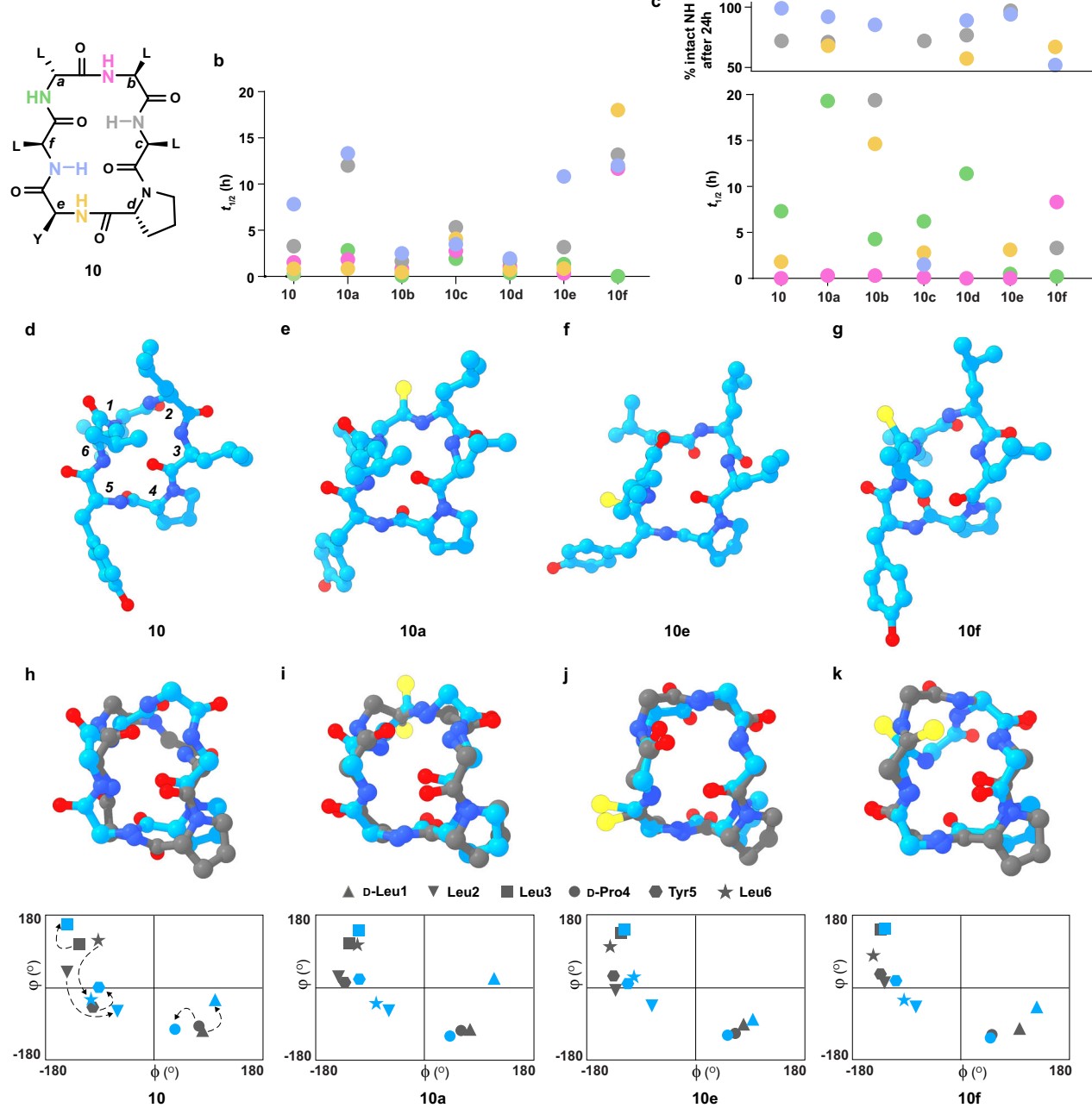

**Fig. 2 | Structural impact of thioamidation on macrocyclic peptides. a** Chemical structure of **10**, with each amide proton color coded. The small letters denote the site of thioamidation. **b** Dot plots representing the half-lives ($t_{1/2}$) of the individual amide protons of **10**, **10a–f** deduced from the HDX experiment in 20% $D_2O$ in DMSO-$d_6$, and **c** in 20% $CD_3OD$ in $CDCl_3$. The slow exchanging amide protons whose $t_{1/2}$ could not be determined have been represented as % remaining after 24 hours. The average solution structure determined by 200 ns fMD simulation of (**d**) **10**, (**e**)

**10a**, (**f**) **10e**, and (**g**) **10f** in DMSO-$d_6$. The amino acid residue numbers are shown only in **10**. The backbone superimposed average structures obtained from the 200 ns rMD (iron) and fMD (aqua) simulation of (**h**) **10**, (**i**) **10a**, (**j**) **10e**, and (**k**) **10f**. The structures are generated using ChimeraX. The Ramachandran plot shows the deviation in torsion angles (ϕ,ψ) of the residues in the average structures obtained from the rMD and fMD simulation by the broken arrows. Source data are provided as a Source Data file.

## Structural impact of thioamidation

To understand the conformational impact of thioamide substitution, the solution structures of **10**, **10a**, **10e**, and **10f** in DMSO-$d_6$ was determined through restrained molecular dynamics (rMD) simulation by using the interproton distances derived from the ROESY spectrum (Supplementary Fig. 6). An unrestrained MD (fMD) simulation was also performed to assess the similarity of the structures (Fig. 2d–g) to the ones derived from experimental restraints. Thioamide substitution either at an externally oriented (**10a**, **10e**) or internally oriented C=O (**10f**) did not introduce a cis-peptide bond, unlike N-methylation[34]. The

RMSD calculated over Cα atoms within **10** and **10a** (0.71 Å), **10e** (0.85 Å), **10f** (0.41 Å) suggests that there are no major overall structural change on thioamidation (Supplementary Figs. 7, 8). However, we note that thioamide modulates the local conformation within the macrocyclic peptide through steric effects.

**10** shows a structurally rigid type II' β turn about D-Pro4$^{i+1}$-Tyr5$^{i+2}$ (Fig. 2d) that is stabilized by Leu3CO•••HNLeu6 H-bond, resulting in slow exchange of Leu6NH (Fig. 2b). On the contrary, the β-turn about D-Leu1$^{i+1}$-Leu2$^{i+2}$ is distorted due to the outward flip of Leu6CO, perhaps to pack the side chains of Leu6 and Leu2. This results in the lack of

Leu6CO•••HNLeu3 H-bond causing rapid exchange of Leu3NH (Fig. 2b). The thioamide substitution at D-Leu1 (**10a**) results in an upward flip of the C=S to avoid the steric clash between S and D-Leu1C$^\beta$ (Fig. 2e). Consequently, we note an alteration in the backbone torsion angles ($\phi$,$\psi$) of D-Leu1 and Leu2 that are associated with the thioamide bond (Supplementary Table 3)[35]. The C=S orientation also alters the $\chi_1$ of Leu2 to a *trans* conformation (as opposed to *gauche*$^+$ in **10**) to avoid steric clash between S and the isopropyl group. This ultimately creates a hydrophobic patch formed by Leu2 and Leu3 side chains around Leu3NH (Fig. 2e), resulting in its slow exchange (Fig. 2b). The thioamide incorporation at Tyr5 in **10e**, forces the Leu6CO inwards to avoid the steric clash between C(S) and Leu6C(O) resulting in a type II' β-turn about D-Leu1$^{i+1}$-Leu2$^{i+2}$ (Fig. 2f and Supplementary Table 3). However, the rapid exchange of Leu3NH suggests the absence of an intramolecular H-bond stabilising the β-turn, in contrast to the β-turn at D-Pro4$^{i+1}$-Tyr5$^{i+2}$ stabilized by Leu3CO•••HNLeu6 H-bond (Fig. 2b). Additionally, the $\chi_1$ of Tyr5 is in a *trans* conformation instead of *gauche*$^+$ in **10**, perhaps to optimize the hydrophobic packing within the aromatic ring and S. Thioamidation at Leu6 (**10f**), results in a structure with maximal resemblance to **10** (Fig. 2g). The local desolvation at the C(S) and its hydrophobic interaction[27] with Leu2 side chain results in solvent shielding of Leu2NH (Fig. 2b). In addition, the unusually slow exchange of Tyr5NH presumably results from the formation of an intramolecular H-bond with Leu3CO resulting in a γ-turn.

To further assess the influence of thioamidation on the conformational flexibility of **10**, the average structures resulting from the restrained and free MD simulations were compared. We note large alterations in the backbone torsion angles of **10** within the two structures, including the residues at the type II' β-turn (D-Pro4 and Tyr5) (Fig. 2h). However, **10a**, **10e**, and **10f** shows low variation in the torsion angles of Leu3, D-Pro4, and Tyr5 as compared to **10** (Fig. 2i–k). This result suggests that thioamide substitution can induce segmental rigidity into macrocyclic peptides. The reduced conformational flexibility on thioamidation was also indicated by a reduction in the spread of the dihedral angles $\phi$ and $\psi$ during the free MD simulation in both polar and nonpolar solvents (Supplementary Figs. 9–11). Therefore, the restricted conformational freedom resulting in enhanced solvent shielding of the amide protons contribute to the improved passive permeability of the macrocyclic peptide on thioamidation[36].

## Desolvation induced by thioamidation drives membrane permeability

Since reduced solvation of S also contributes to the lipophilicity of the thioamidated peptides, we sought to estimate the global desolvation of the macrocyclic peptides, which is affected by solvation of both N(H) and C(S). Utilizing a series of N-methylated peptides, Burton and colleagues previously showed that the primary determinant of cellular permeability is the desolvation potential of the polar functional groups in the peptide, followed by its lipophilicity[37]. Thus, we evaluated the desolvation potential of the cyclic peptides by determining the distribution coefficients in heptane-ethylene glycol (logD$_{h/e}$), a solvent system that provides an experimental estimate of desolvation or hydrogen bond potential[38]. Gratifyingly, the most permeable cyclic peptides **10a**, **10e**, and **10f** showed higher distribution coefficients indicating lower desolvation penalty than the rest (Fig. 3a). Consequently, an improved correlation was observed between PAMPA permeability and desolvation penalty (Fig. 3b) over lipophilicity (Supplementary Fig. 12). This supports the hypothesis that reduced desolvation of the amide bonds directly induced by thioamidation and indirectly through increased solvent shielding of the amide protons is the major driver of passive transcellular permeability.

We next determined the permeability of the peptides across confluent monolayers of human adenocarcinoma cells (Caco-2) as a model of the intestinal epithelium. **10** showed low permeability (0.76 × 10$^{-6}$ cm/s) in the apical to basolateral (A-B) direction and high

permeability (12.4 × 10$^{-6}$ cm/s) in the basolateral to apical (B-A) direction, suggesting the involvement of efflux pumps in lowering the permeability of **10** (Fig. 3c). **10a**, **10e**, and **10f** also showed high efflux ratio (Supplementary Table 4), however, they displayed enhanced permeability than **10** except **10c**, which showed rapid exchange of amide protons (Fig. 2b). When the efflux pump was inhibited[39] (confirmed by an efflux ratio of <2), a significant improvement in the permeability of the peptides was observed that correlated with the values obtained from the PAMPA (Fig. 3d)[40]. These results suggest that thioamide substitution improves the passive permeability of cyclic peptides across the lipid bilayer in live cells. It is also worth noting that **10a**, **10e**, and **10f** show higher permeability than the orally bioavailable atenolol in the absence of efflux pump inhibitor (Fig. 3c).

## Comparison of thioamidation and N-methylation

To evaluate the performance of thioamidation in improving the permeability of macrocyclic peptides against the most dominant modification, N-methylation, we synthesized the N-methylated analogs of **10** (**10g–10k**) (Supplementary Table 1). N-methylation enhanced the lipophilicity of **10**, as observed from the increased C18 retention time, reduced solubility, and increased logD$_{7.4}$ of **10g–10k** (Fig. 4a and Supplementary Fig. 13a, b). Except for the amide bond 1 A, thioamidation resulted in a more pronounced effect than N-methylation at all other sites. Subsequently, the N-methylated analogs showed comparable passive permeability to their thioamidated counterparts, except at amide bond 6 A, where the thioamidated analog showed better permeability (Fig. 4b). Curiously, either thioamidation or N-methylation of amide bond 1 A followed by 5 A resulted in the most permeable analogs within the series.

Macrocyclic peptides are endowed with proteolytic stability associated with their restricted conformational freedom; however, we were keen to evaluate their stability under the harsh proteolytic condition in the gastrointestinal tract[41]. Despite the presence of two D-amino acid residues (D-Leu1 and D-Pro4) that are usually considered to confer proteolytic resistance to peptides, **10** underwent rapid degradation in Simulated Gastric Fluid (SGF) ($t_{1/2}$ 12 min), consisting of only pepsin at pH 1.2[42] (Fig. 4d and Supplementary Fig. 14). Chemical modification of an amide bond targeted by a protease increases its half-life[43,44]. Thus, a significant increase in the half-life of the peptides by both thioamidation and N-methylation at the amide bonds 1 A and 5 A indicate that 1 A and 5 A are the direct cleavage sites of pepsin on the macrocycle, with 5 A (flanked by Tyr5 and Leu6) being the primary cleavage site. Curiously, the increased half-life on thioamidation but not N-methylation at 6 A suggests that it is not the direct cleavage site of pepsin. However, thioamidation at 6 A perhaps slows down the cleavage at the preceding amide bond 5 A and the following amide bond 1 A, as observed earlier for linear peptides treated against serine proteases[43].

We next assessed the stability of the macrocyclic peptides in Simulated Intestinal Fluid (SIF)[42], comprising of pancreatin, which is a mixture of several proteases produced by the exocrine cells of the porcine pancreas. While the parent peptide **10** showed low proteolytic stability ($t_{1/2}$ 16 min), the maximum protection was achieved by thioamidation at D-Leu1 (**10a**) ($t_{1/2}$ 147 min), Tyr5 (**10e**) ($t_{1/2}$ 241 min), and Leu6 (**10f**) ($t_{1/2}$ 126 min) (Fig. 4e and Supplementary Fig. 15), as observed in SGF. In addition to 1 A and 5 A, either chemical modification at 2 A protects the peptide from rapid degradation, highlighting the direct cleavage sites in SIF. In contrast, the increase in half-life only by thioamidation at 4 A and 6 A points at the indirect protection offered by thionating the preceding and following residues of 5 A. Nonetheless, the protection offered by thioamidation outweigh the effect of N-methylation in these type of macrocyclic peptides. The combined results of SGF and SIF highlight the benefit of thioamidating hydrophobic amino acids to protect macrocyclic peptides against proteolysis.

Human liver is the primary site for drug metabolism, decreasing the amount of drug entering the systemic circulation when

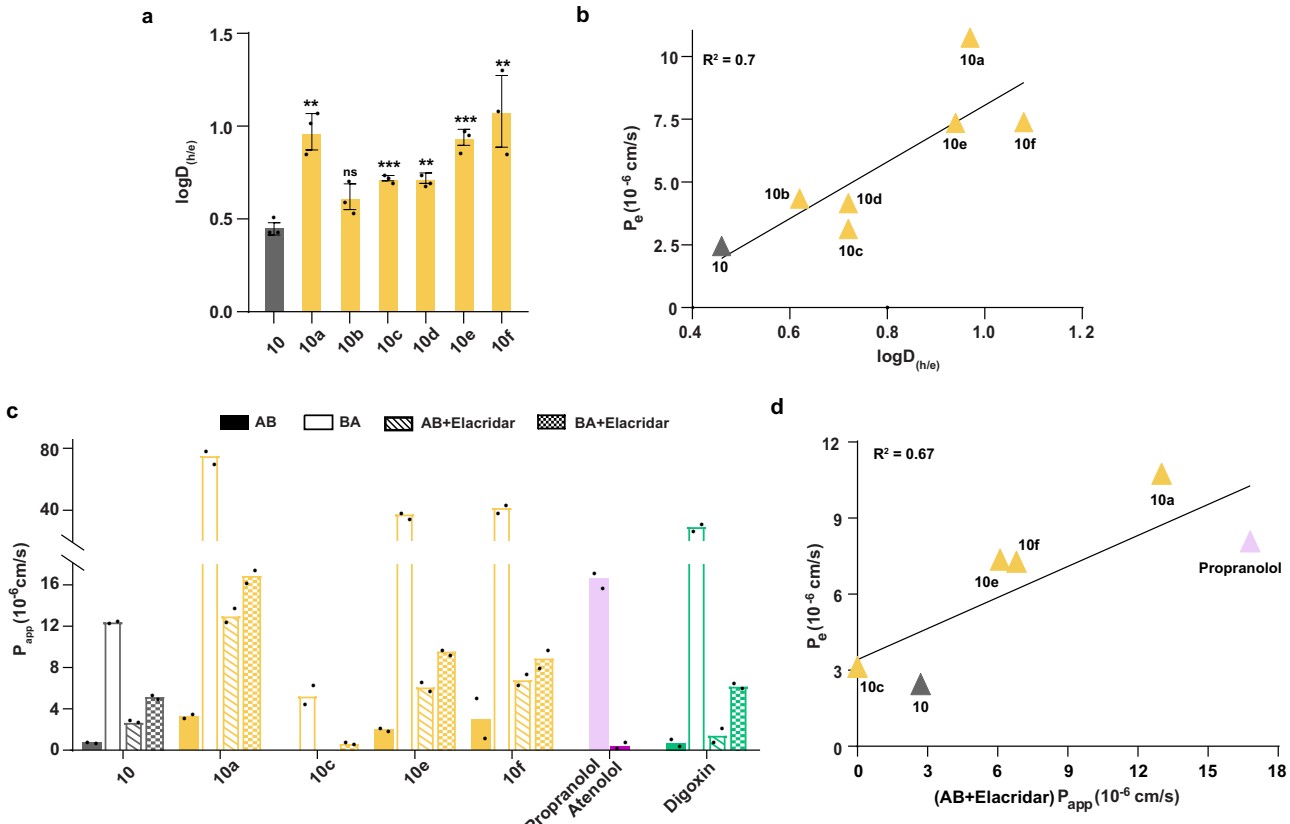

**Fig. 3 | Influence of desolvation on the permeability of macrocyclic peptides.**
**a** Partition coefficient of **10** and **10a–10f** in heptane-ethylene glycol (h/e) solvent
system. $n = 3 \pm$ SEM. **b** Linear correlation plot of $P_{e(PAMPA)}$ vs. $\log D_{(h/e)}$.
**c** Bidirectional Caco-2 permeability in the presence and absence of P-gp efflux
pump inhibitor elacridar, represented in bars as $P_{app}$. Propranolol and Atenolol
were used markers of high and low permeability, respectively. Digoxin was used as
the control substrate of P-gp efflux pumps. $n = 2$. Since $n < 3$, error bars and

associated statistics has not been derived. **d** Linear correlation plot of $P_{e(PAMPA)}$ vs.
(AB + Elacridar) $P_{app(Caco-2)}$. Each bar represents mean values of three biological
replicates: dots are individual data points. Statistical significance was measured
against the (all-amide) parent molecule by a one-tailed unpaired $t$-test. $^*p < 0.05$,
$^{**}p < 0.01$, $^{***}p < 0.001$, $^{****}p < 0.0001$, ns (non-significant) $> 0.05$. $p$ (**10** vs **10a, 10b,
10c, 10d, 10e, 10f**) = 0.0028, 0.0511, 0.0007, 0.0019, 0.0003, 0.0088 for $\log D_{(h/e)}$.
Source data are provided as a Source Data file.

administered through the oral route. Thus, we assessed the in vitro
stability of the macrocyclic peptides against the drug-metabolizing
cytochrome P450 (CYP) enzymes present in the human liver micro-
somes to determine the clearance by the liver (Supplementary Figs. 16
and 13c)[14]. The parent macrocycle, **10**, underwent fast degradation
resulting in its rapid clearance ($CL_{int}$ 225 µL/min/mg), whereas, both
thioamidation and N-methylation prevented the macrocycle from
degradation and lowered its clearance with varying efficacy (Fig. 4f).
We finally assessed the ex vivo stability of the macrocycles in human
blood plasma (Supplementary Fig. 17), which showed two folds or
more enhanced half-life of **10a–f** compared to **10** ($t_{1/2}$ 8 h). The
N-methylated analogs also showed comparable half-lives to the
thioamidated analogs except **10j** and **10k** (Supplementary Fig. 13d).
This led us to assess their binding to plasma proteins (Fig. 4c), since
plasma protein binding is beneficial for extending the circulation half-
life of peptides. The thioamidated analogs **10d–10f** along with the
N-methylated analog **10j** showed significantly higher plasma protein
binding than **10**. This result indicates that compound lipophilicity
induced by thioamidation results in amplified plasma protein binding.

**In vivo pharmacokinetic (PK) assessment of the thio-peptides**
The improved in vitro pharmacokinetic properties of the thioamidated
analogs led us to evaluate the absorption of the macrocyclic peptides
into the systemic circulation post oral administration in male Wistar
rats. **10a, 10e**, and **10f**, which showed enhanced permeability (Fig. 4b)
and improved metabolic stability than the parent macrocycle **10**
(Fig. 4d–f), were chosen for the evaluation (Fig. 5a). In addition, we also

selected **10h** and **10k** to compare the effect of N-methylation versus
thioamidation at the amide bonds 1 A and 5 A (Fig. 5b). Since we sought
to evaluate the sole effect of the chemical modification on the in vivo
PK properties of the macrocyclic peptides, we conducted the i.v.
(intravenous) and p.o. (per oral) studies without using any micro-
emulsion formulation that are known to improve the oral bioavail-
ability of peptides[45]. The low in vivo permeability and metabolic
stability of **10** was evident from its maximum concentration in the
blood ($C_{max}$: 3.5 ng ml$^{-1}$) after a 10 mg/kg oral administration (Table 1).
Furthermore, its plasma exposure rapidly declined, reaching an
undetectable limit by 1 h. On the contrary, **10a, 10e**, and **10f** achieved a
$C_{max}$ of 15 ng ml$^{-1}$, 47.2 ng ml$^{-1}$, and 61.7 ng ml$^{-1}$, respectively, post oral
administration and could be detected till 8 h or more (Fig. 5a). The
stark improvement in the plasma exposure of **10a** (26.9 ng h ml$^{-1}$; ~ 30
folds), **10e** (359.5 ng h ml$^{-1}$; ~ 400 folds), and **10f** (120.8 ng h ml$^{-1}$; ~ 130
folds) with respect to **10** (0.9 ng h ml$^{-1}$) till the time of the last mea-
surable concentration ($AUC_{last}$) emphasize the contribution of this
single atom substitution in improving the bioavailability of macro-
cyclic peptides (Table 1). Remarkably, the N-methylated analogs **10h**
and **10k** show comparable $C_{max}$ and plasma exposure (vital parameters
for determining the drug exposure) to their respective thioamidated
analogs post oral administration (Table 1 and Supplementary Table 5).
Although, we note a higher fraction of **10k** reaching the systemic cir-
culation than **10e** by the oral route (represented by %F), the compar-
able enhancement in their plasma exposure post oral administration
with respect to **10**, suggest the potential benefit of thioamidation in
improving the in vivo PK of macrocyclic peptides.

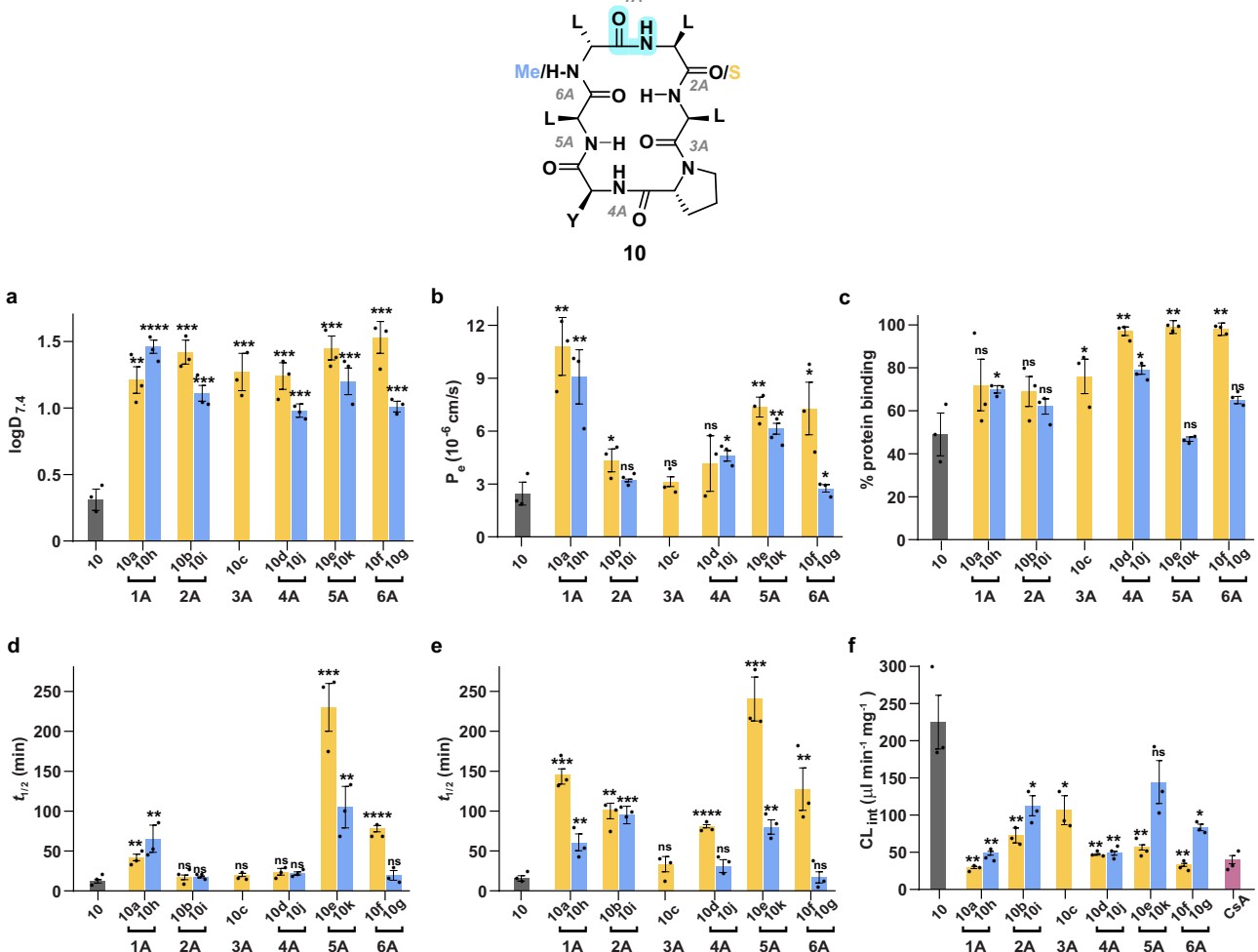

**Fig. 4 | Comparison of HBA and HBD-masking on the in vitro PK properties.** 1A-6A refer to the amide bonds in **10**. The yellow and the blue bars in the individual panels represent the thioamidated (**10a**–**10f**) and the N-methylated (**10g**–**10k**) counterparts, respectively, at a specific amide bond. **a** Octanol-water partition coefficient ($logD_{7.4}$). **b** Membrane permeability ($P_e$) of the macrocyclic peptides determined by the PAMPA. **c** Extent of the peptides to bind non-specifically to plasma proteins determined through Rapid Equilibrium Dialysis (RED) represented in percentage (%). Half-lives ($t_{1/2}$) of the peptides in **d** simulated gastric fluid (SGF), and **e** simulated intestinal fluid (SIF). **f** Intrinsic clearance ($CL_{int}$) of the peptides in human liver microsomes deduced from their $t_{1/2}$ values. Cyclosporin A (CsA) was taken as a control ($CL_{int}$: $40 \pm 5.5$ µL min$^{-1}$ mg$^{-1}$). $n = 3 \pm$ SEM. Each bar represents mean values of three biological replicates: dots are individual data points. Statistical significance was measured against the (all-amide) parent molecule by a one-tailed unpaired $t$-test. *$p < 0.05$, **$p < 0.01$, ***$p < 0.001$, ****$p < 0.0001$, ns (non-

significant) $> 0.05$. $p$ (**10** vs **10a, 10b, 10c, 10d, 10e, 10f**) = 0.0013, 0.0002, 0.0007, 0.0005, 0.0002, 0.0003; $p$ (**10** vs **10g, 10h, 10i, 10j, 10k**) = 0.0003, <0.0001, 0.0004, 0.0005, 0.0008 for $logD_{7.4}$. $p$ (**10** vs **10a, 10b, 10c, 10d, 10e, 10f**) = 0.0024, 0.0449, 0.1704, 0.1228, 0.0015, 0.0194; $p$ (**10** vs **10g, 10h, 10i, 10j, 10k**) = 0.3501, 0.0065, 0.1232, 0.0102, 0.0021 for $P_e$ by PAMPA. $p$ (**10** vs **10a, 10b, 10c, 10d, 10e, 10f**) = 0.1022, 0.0802, 0.0418, 0.0032, 0.0026, 0.0026; $p$ (**10** vs **10g, 10h, 10i, 10j, 10k**) = 0.0802, 0.0421, 0.1372, 0.0158, 0.3868 for plasma-protein binding. $p$ (**10** vs **10a, 10b, 10c, 10d, 10e, 10f**) = 0.0012, 0.1085, 0.168, 0.107, 0.0008, <0.0001; $p$ (**10** vs **10g, 10h, 10i, 10j, 10k**) = 0.1142, 0.0095, 0.245, 0.113, 0.0045 for $t_{1/2}$ in SGF. $p$ (**10** vs **10a, 10b, 10c, 10d, 10e, 10f**) = 0.0002, 0.0023, 0.0509, <0.0001, 0.0007, 0.0080; $p$ (**10** vs **10g, 10h, 10i, 10j, 10k**) = 0.2716, 0.0081, 0.0001, 0.324, 0.0095 for $t_{1/2}$ in SIF. $p$ (**10** vs **10a, 10b, 10c, 10d, 10e, 10f**) = 0.0033, 0.0080, 0.0246, 0.0046, 0.0056, 0.0035; $p$ (**10** vs **10g, 10h, 10i, 10j, 10k**) = 0.0100, 0.0048, 0.0241, 0.0048, 0.0801 for $CL_{int}$. Source data are provided as a Source Data file.

## Extension of hypothesis on bioactive macrocycle

Encouraged by the significant improvement in plasma concentration of the thioamidated model macrocyclic peptides post oral administration, we sought to evaluate the therapeutic benefit of this single-atom substitution. The macrocyclic Somatostatin agonist cyclo(-D-Trp[1]-Lys-Thr-Phe-Pro-Phe[6]-) (**13**) was chosen to investigate the impact of thioamidation on the bioactivity of the macrocyclic peptide in vivo (Fig. 6a)[46]. The fundamental difference between the parent macrocycles **10** and **13** is their lipophilicity. As opposed to **10** with a $logD_{7.4}$ of 0.31, the highly solvated structure of **13** ($logD_{7.4} = -1.2$) is evident from the rapid exchange of individual amide protons during the HDX experiment ($t_{1/2}$ for all H$^N$s are < 5 min) (Supplementary Fig. 18).

Since polar substitution on macrocyclic peptides is known to tremendously impede their passive permeability[47], we thus sought to evaluate the lipophilicity (Supplementary Fig. 19) and membrane

passage on thioamidating **13**. On thioamidation, a lower desolvation penalty was observed for several analogs (Fig. 6b) that resulted in a modest permeability enhancement of **13a**, **13b**, and **13d** in the PAMPA (Fig. 6c), where the O to S substitutions were present at the solvent-exposed C=O (Fig. 6a). However, the most permeable thioamidated analog, **13b**, showed comparable permeability to the lipophilic macrocycle **10**. Consequently, **13** and **13a**–**13f** could not permeate the Caco-2 monolayer (Supplementary Table 6). This emphasizes the challenge of achieving passive transcellular permeability of macrocyclic peptides with polar amino acid side chains, on monothioamidation.

## Prolonged plasma exposure of thio-peptides following s.c. injection

With the failed attempt to improve the intestinal permeability of the polar macrocycle **13** by single atom substitution, we focused on

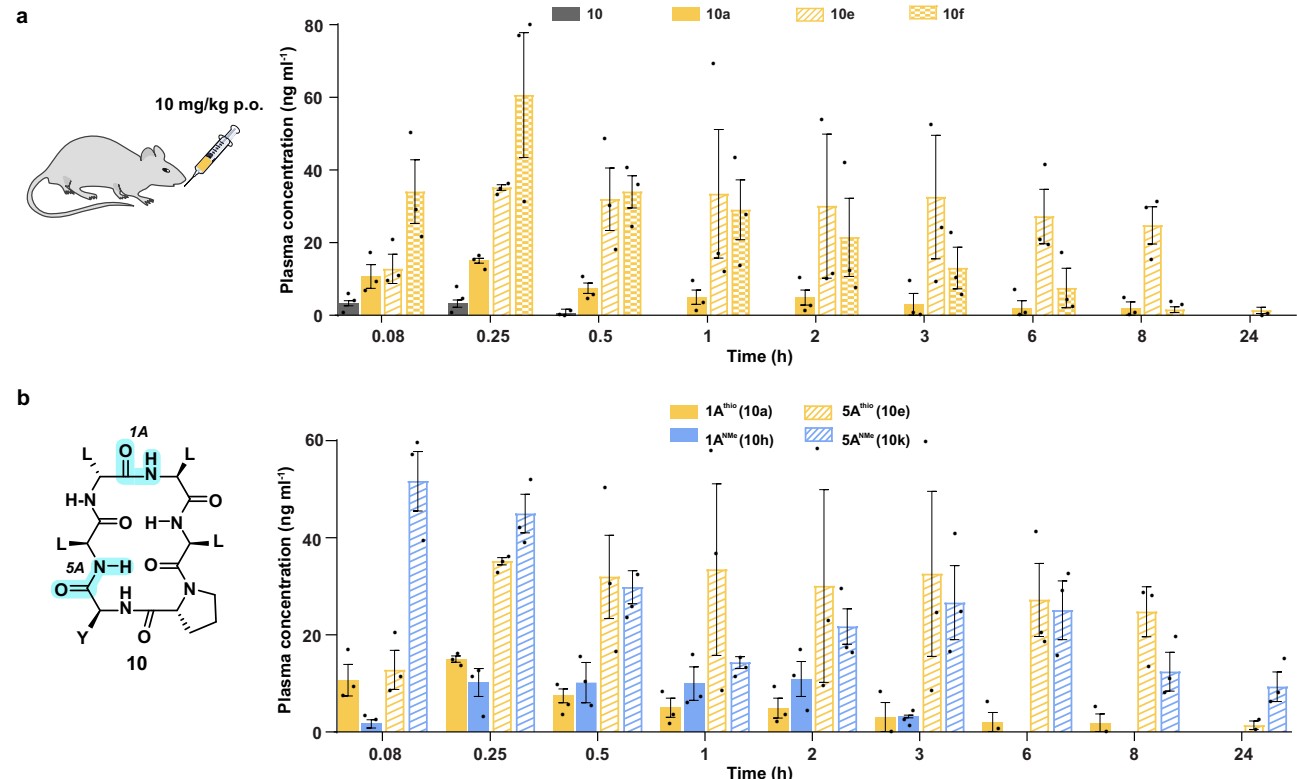

**Fig. 5 | Plasma exposure of thioamidated and N-methylated macrocyclic peptides post oral administration. a** Plasma concentration vs time plot of **10, 10a, 10e,** and **10f** after a single dose oral administration (10 mg/Kg) in male Wistar rats. $n = 3 \pm$ SEM. **b** Comparison of the plasma concentration with respect to time of the macrocycles obtained by thioamidation and N-methylation of amide bonds 1 A

(**10a, 10h**) and 5 A (**10e, 10k**), following a single dose oral administration (10 mg/Kg) in male Wistar rats. $n = 3 \pm$ SEM. Each bar represents mean values of three biological replicates: dots are individual data points. Source data are provided as a Source Data file.

### Table 1 | In vivo pharmacokinetics in male Wistar rats

| Peptides | | p.o | | | | i.v | | | | | %F |
|---|---|---|---|---|---|---|---|---|---|---|---|
| | | $C_{max}$ (ng ml$^{-1}$) | $MRT_{last}$ (h) | $AUC_{last}$ (ng h ml$^{-1}$) | Fold Change | $C_{max}$ (ng ml$^{-1}$) | $MRT_{last}$ (h) | $V_{ss}$ (l kg$^{-1}$) | $CL_{obs}$ (ml min$^{-1}$ kg$^{-1}$) | $AUC_{last}$ (ng h ml$^{-1}$) | |
| **10** | | 3.5 | 0.2 | 0.9 | - | 1046.7 | 1.6 | 4.1 | 28.4 | 548.1 | 0 |
| **1 A** | **10a** | 15.0 | 2.0 | 26.9 | 31 | 48.1 | 2.8 | 28.6 | 91.1 | 146.4 | 1.8 |
| | **10h** | 13.1 | 1.3 | 25.9 | 29 | 434.6 | 0.4 | 2.3 | 79.5 | 212.2 | 1.2 |
| **5 A** | **10e** | 47.2 | 5.9 | 359.5 | 409 | 2660.9 | 1.5 | 0.7 | 7.5 | 2278.4 | 1.6 |
| | **10k** | 55.9 | 9.0 | 344.9 | 393 | 1165.5 | 0.4 | 0.8 | 33.3 | 509.5 | 6.8 |
| **10f** | | 61.7 | 2.1 | 120.8 | 138 | 580 | 1.1 | 3.2 | 42.2 | 389.7 | 3.1 |

$n = 3$. For clarity, only the mean is presented; $n = 3 \pm$ SEM is given in Supplementary Table 5. Each value represents mean of three biological replicates. Statistical significance was measured by a one-tailed unpaired $t$-test. \**p < 0.05*, \*\**p < 0.01*, \*\*\**p < 0.001*, \*\*\*\**p < 0.0001*. %F = $(AUC_{last(p.o.)}/AUC_{last(i.v.)})$ x $(100/x)$, x = p.o./i.v. dose. $p$ (**10** vs **10a, 10h, 10e, 10k, 10f**) = \* (0.0276), \* (0.0382), \*\* (0.0071), \*\* (0.0048), \* (0.0371) for $AUC_{last}$ (p.o.).

subcutaneous (s.c.) injection, which is the most common route of therapeutic peptide administration in the development pipeline[48]. Wherein the major challenge that needs to be addressed is their rapid clearance from plasma owing to fast enzymatic degradation and renal clearance attributing to short half-lives[49]. Thus, we tested the stability of the thioamidated macrocycles **13a**–**13f** in plasma (Fig. 7a). As observed for the hydrophobic macrocycle **10** (Supplementary Fig. 17), a single amide to thioamide substitution in most cases increased the half-life of **13** ($t_{1/2}$ 15 h) by two folds or more. Likewise, we observe a significant increase in the binding to plasma proteins by **13a**–**13f** (Fig. 7b), which correlates with their increased lipophilicity (logD$_{7.4}$) (Supplementary Fig. 19c). Subsequently, we selected **13a**, **13b**, **13d**, and **13e** for assessing their plasma exposure post single dose subcutaneous injection into male Sprague Dawley rats. The

parent macrocycle **13** was efficiently absorbed into plasma; however, its concentration rapidly dropped within 2 h (8 ng ml$^{-1}$). On the contrary, **13a** (40 ng ml$^{-1}$), **13b** (29 ng ml$^{-1}$), **13d** (17 ng ml$^{-1}$), and **13e** (39 ng ml$^{-1}$) showed higher plasma concentration at 2 h and were slowly absorbed into the plasma than **13** (Fig. 7c and Supplementary Table 7). Thus, the enhanced lipophilicity of the macrocyclic peptides by thioamidation result in higher binding to the plasma membrane and/or with proteins like albumin at the site of injection. This leads to their reduced diffusion rate and longer retention at the site of injection.

### In vitro efficacy and docking with SSTR2
Next, we sought to assess the in vivo efficacy of the thioamidated somatostatin analogs post s.c. administration. Somatostatin, first

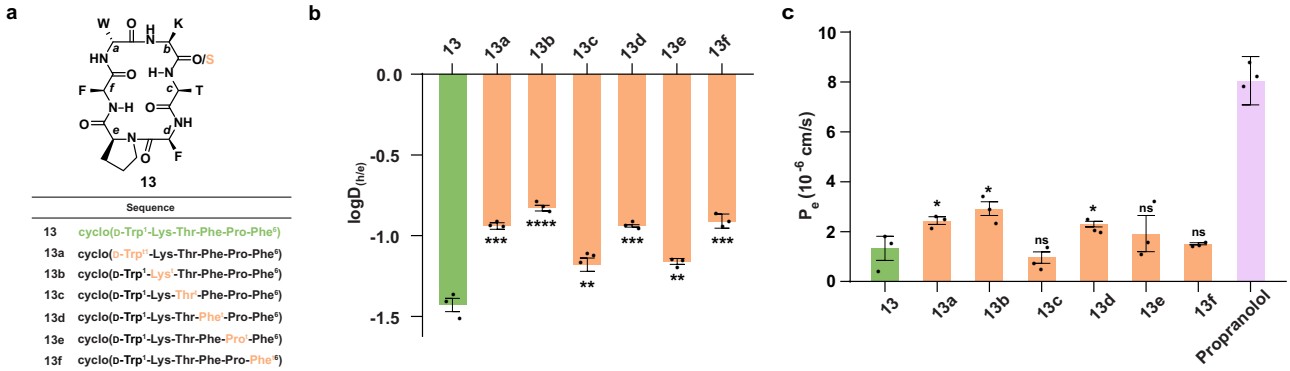

**Fig. 6 | Impact of thioamide on hydrophilic macrocyclic peptide. a** The sequence of the cyclic hexapeptide analog of Somatostatin (**13**) and its mono-thioamidated regioisomers (**13a–13f**). **b** Partition coefficient of **13**, **13a–13f** in heptane-ethylene glycol (h/e) solvent system indicating their desolvation penalty. **c** Membrane permeability of the macrocyclic peptides determined by the PAMPA ($P_e$). Propranolol was used as the marker for transcellular transport. $n = 3 \pm$ SEM. Each bar represents mean values of three biological replicates: dots are individual data points. Statistical significance was measured against the (all-amide) parent molecule by a one-tailed unpaired $t$-test. $*p < 0.05$, $**p < 0.01$, $***p < 0.001$, $****p < 0.0001$, ns (non-significant) $> 0.05$. $p$ (**13** vs **13a, 13b, 13c, 13d, 13e, 13f**) = 0.0002, <0.0001, 0.0052, 0.0002, 0.0020, 0.0005 for $logD_{(h/e)}$. $p$ (**13** vs **13a, 13b, 13c, 13d, 13e, 13f**) = 0.0347, 0.0183, 0.2371, 0.0472, 0.2610, 0.3540 for $P_e$ by PAMPA. Source data are provided as a Source Data file.

known to inhibit the release of growth hormone, induces its biological effects by interacting with five of its G-protein-coupled receptor subtypes SSTR1-SSTR5[50]. Out of which, the SSTR2 and SSTR5 have gained significant attention due to their role in mediating the inhibition of growth hormone and antiproliferative effects of somatostatin on tumor growth. The lack of therapeutic effectiveness of native somatostatin lies in its short plasma half-life (~3 min), and unselective binding to all the receptor subtypes SSTR1-5. We, thus, assessed the selectivity of **13** and the thioamidated analogs against SSTR2 and SSTR5. **13**, **13b**, and **13e** were found to be selective against SSTR2 over SSTR5, while **13d** showed exquisite specificity against SSTR2 (Fig. 7d and Supplementary Fig. 20). A complete loss in binding against SSTR2 and SSTR5 was observed for **13a, 13c**, and **13f**. To understand the structural basis for receptor binding, we determined the solution structure of **13** and **13a–13f** using interproton distances derived from 2D NMR (Supplementary Fig. 21). These NMR-derived structures of the peptides were then docked into the recently determined structure of SSTR2 bound to the FDA-approved drug Octreotide (SRIF-14; PDB code: 7T11)[51] using rigid receptor docking approach.

The parent peptide **13** is predicted to bind to SSTR2 at the bottom of the ligand binding pocket with a pose comparable to Octreotide, where several key interactions between the pharmacophore of **13** (-Phe-D-Trp-Lys-Thr-) and SSTR2 retain the ligand in its position[51,52]. In particular, the indole ring of D-Trp1 inserts within a hydrophobic pocket and engages in an edge-to-face π-stacking interaction with Phe208[5.38]. Additionally, the carbonyl oxygen of D-Trp forms a H-bond with the side chain of Asn276[6.55]. The εNH₂ of Lys2 engages in a salt bridge with Asp122[3.32], H-bond with Gln126[3.36], and a cation-π interaction with Tyr302[7.43]. Lastly, the Phe6 interacts with Tyr205[5.35] via a π-π stacking interaction and Thr3 OH forms a H-bond with Gln102[2.63] (Supplementary Fig. 22). All these interactions were retained in **13b, 13d**, and **13e** (Supplementary Fig. 23). In contrast, a complete loss of these interactions was observed in the inactive analogs **13c** and **13f**, which showed significant alteration in their native structure with respect to **13** due to an O to S substitution at an internally oriented carbonyl group (Supplementary Fig. 24b, c). However, despite the comparable solution structure of **13** with **13a** having a solvent-exposed D-Trp1 C=S, the loss of its binding to SSTR2 demonstrates the crucial role of a single H-bond (D-Trp1CO-Asn276NH₂[δ]) in ligand binding which is abrogated on thioamidation (Supplementary Fig. 24a).

**In vivo inhibition of GH.** Finally, we estimated the in vivo efficacy of **13** and the thioamidated analogs **13b, 13d**, and **13e** in suppressing the release of growth hormone (GH) post subcutaneous injection at 250 μg/kg dosage (Fig. 7e). **13a** was also included as a negative control to assess its effect on GH release. The rats were bled from their orbital sinus, and the plasma GH levels were evaluated through sandwich ELISA. Except **13a**, we observed a significant inhibition of unstimulated GH release by **13, 13b, 13d**, and **13e**. Remarkably, despite the strongest inhibition of GH release at the initial time point (1 h) by **13**, the thioamidated analogs showed better inhibition of GH release at the subsequent time points throughout the course of experiment over 48 h (Fig. 7e). This correlates well with the high plasma concentration of **13** at the initial time points and a rapid drop within 2 h of s.c. administration (Fig. 7c). **13** has the least potency in inhibiting the GH release over 0-48 h (Fig. 7f), wherein it shows substantial inhibition only within 24 h (Fig. 7g). In contrast, the inhibitory effect lasts longer in **13b, 13d**, and **13e** showing a significant reduction in plasma GH levels within 24–48 h (Fig. 7h). **13d** shows the maximum inhibition of GH release, which is related to its longer retention in the plasma (Fig. 7c) due to its high plasma stability (Fig. 7a) than the other thioamidated analogs. The rapid drop in plasma concentration of **13** is responsible for its least in vivo efficacy. In comparison, the increased metabolic stability and lipophilicity of the macrocyclic peptide on thioamidation resulted in long-acting analogs that slowly diffused into plasma from the site of injection.

## Discussion

Desolvation of peptide bonds by targeting the hydrogen bond donor (NH groups) through conformational control, steric occlusion, and N-methylation remain the state-of-the-art method to improve the membrane permeability of macrocyclic peptides. In the pursuit of an alternative chemical strategy to desolvate the peptide bond with minimal perturbation to the macrocycle conformation, we explored thioamidation. Our results show that masking the hydrogen bond acceptor >C=O with >C=S is effective in improving the passive membrane permeability of macrocyclic peptides composed of hydrophobic amino acid side chains. We note that lipophilicity of the parent macrocycle on which the thio-scan is performed, contributes largely to the absolute permeability of the thioamidated analogs (Fig. 1f). Furthermore, despite identical AlogP of the thioamidated regioisomers **10a–10f, 11a–11g**, and **12a–12h**, they differ in passive permeability across the membrane. The improved permeability of the thioamidated

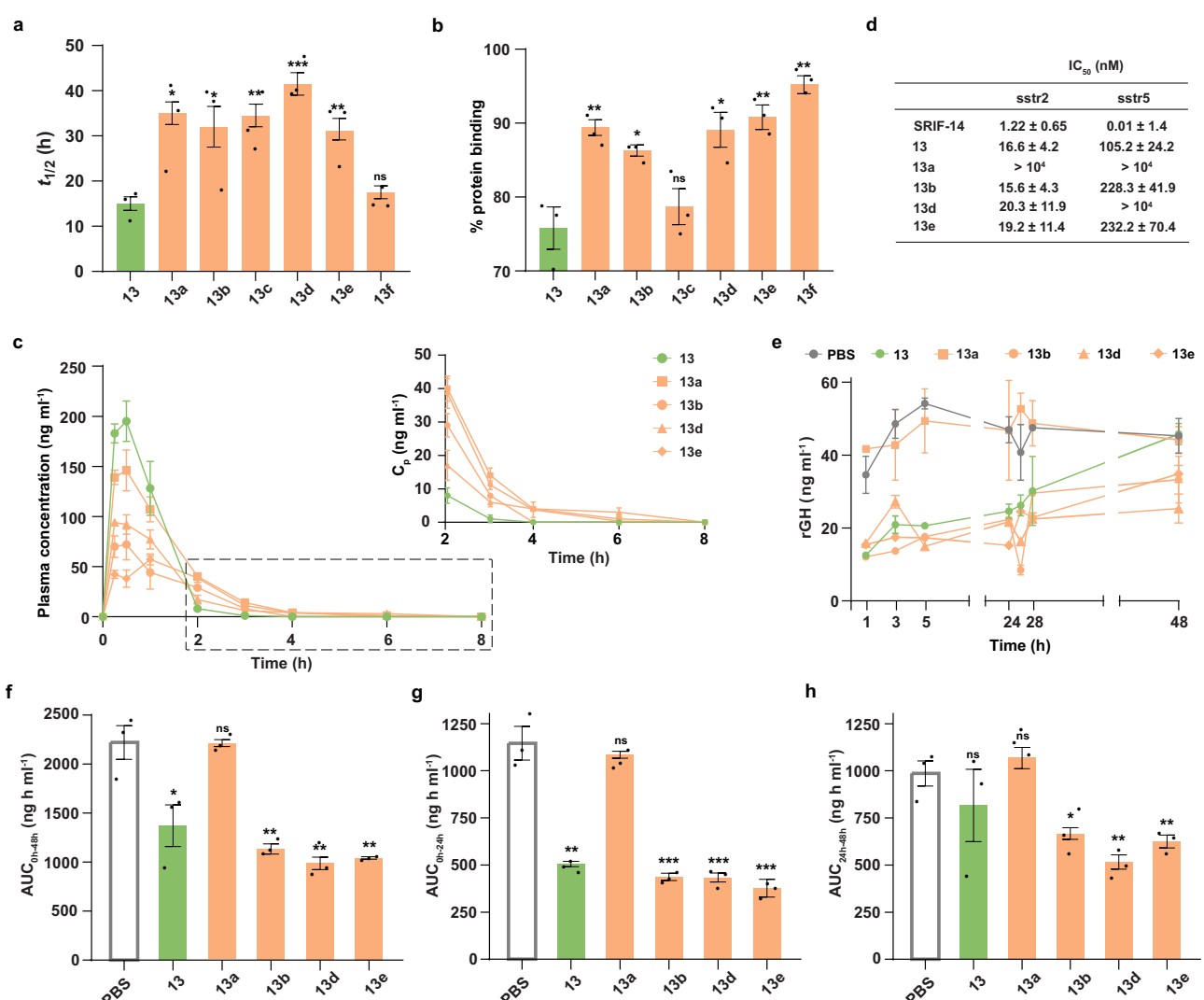

**Fig. 7 | Effect of thioamide on the PK and in vivo efficacy of somatostatin analogs. a** Half-lives ($t_{1/2}$) of **13**, **13a–13f** determined in human blood plasma. **b** % protein binding of the peptides in plasma, as measured through RED. **c** Plasma concentration vs time plot of **13**, **13a**, **13b**, **13d**, and **13e** after a single dose (250 μg/Kg) s.c. administration in male Sprague-Dawley rats. The rapid drop in the concentration of **13** as opposed to its four thio analogs in plasma at later time points (2-8 h) are shown in the inset. **d** The $IC_{50}$ values of the peptides determined from dose-dependent inhibition of forskolin-induced cAMP production in HEK293T cells expressing SSTR2 and SSTR5. $n = 3 \pm$ SEM. **e** Inhibition of rat growth hormone (rGH) levels over 48 hours post single dose (250 μg/Kg) peptide (**13**, **13a**, **13b**, **13d**, **13e**) and PBS (Untreated) administration via the s.c. route. **f** Bar plots represent the area under the concentration-time curve (AUCs) of rGH for (**f**) 0–48 h, (**g**) 0–24 h, and

(**h**) 24-48 h. $n = 3 \pm$ SEM (**a** to **d**); $n = 3 \pm$ SEM (**e**); $n = 3 \pm$ SEM (**f** to **h**), Each bar represents mean values of three biological replicates: dots are individual data points. Statistical significance was measured against the (all-amide) parent molecule by a one-tailed unpaired $t$-test. *$p < 0.05$, **$p < 0.01$, ***$p < 0.001$, ****$p < 0.0001$, ns (non-significant) $> 0.05$. $p$ (**13** vs **13a, 13b, 13c, 13d, 13e, 13f**) = 0.0110, 0.0332, 0.0037, 0.0005, 0.0083, 0.1674 for $t_{1/2}$ in human blood plasma. $p$ (**13** vs **13a, 13b, 13c, 13d, 13e, 13f**) = 0.0051, 0.0109, 0.2305, 0.0124, 0.0050, 0.0016 for plasma-protein binding. For $AUC_{0h-48h}$, $p$ (**PBS** vs **13, 13a, 13b, 13d, 13e**) = 0.0189, 0.3960, 0.0022, 0.0015, 0.0014. For $AUC_{0h-24h}$, $p$ (**PBS** vs **13, 13a, 13b, 13d, 13e**) = 0.0010, 0.2861, 0.0007, 0.0007, 0.0005. For $AUC_{24h-48h}$, $p$ (**PBS** vs **13, 13a, 13b, 13d, 13e**) = 0.2236, 0.0552, 0.0113, 0.0018, 0.0041. Source data are provided as a Source Data file.

macrocycles through artificial (PAMPA) and biological (Caco-2 cells) membrane arises from a concerted effect of conformational restriction, formation of intramolecular H-bonds, and local desolvation of the thioamide bond.

The comparative study of mono-thioamidation versus mono-N-methylation on the macrocycle **10** suggests that both HBA and HBD masking is beneficial in improving the passive membrane permeability of macrocyclic peptides. Our results also show that thioamidation provides metabolic stability to macrocyclic peptides against degradation by the harsh enzymes present in the GI tract. We note that thioamidation at the proteolytically labile peptide bond or at the preceding/following residues confer protection against proteolysis (Fig. 4d, e). This was further verified by performing the metabolite analysis of **13** post SIF treatment, where we identified -Lys2-Thr3-

peptide bond as the primary cleavage site resulting in $t_{1/2}$ of 1.8 h (Supplementary Fig. 25). Thioamidation at the cleavage (P1) site (Lys[2], **13b**) and the preceding (P2) site (D-Trp1, **13a**) prevented the degradation of the macrocycle over 72 h, while thioamidation at the P1' site (Thr[3], **13c**) did not significantly increase the half-life of **13c** ($t_{1/2}$ 2.5 h) (Supplementary Fig. 25).

The marked improvement in permeability and metabolic stability of **10a**, **10e**, and **10f** over **10** led us to perform the complete in vivo PK profiling along with their N-methylated counterparts **10h** and **10k** for comparison. **10a** and **10h**, despite being the most permeable thioamidated and N-methylated analogs, respectively, showed the least plasma exposure post oral administration. Whereas, **10e** and **10k** with moderate permeability and higher metabolic stability (than **10a** and **10h**) against the proteases in the GI tract showed high plasma

exposure post oral administration. **10f** with lower stability than **10e** but with comparable membrane permeability displayed lower plasma exposure than **10e**. This highlights the importance of achieving high proteolytic stability of macrocyclic peptides in the GI tract addition to passive transcellular permeability, to obtain improved plasma exposure post oral administration.

Peptide macrocycles that show high solvent exposure of the backbone amide bonds due to the presence of polar side chains (**13**) failed to attain Caco-2 permeability on single O to S substitution. Nevertheless, the desolvation of the macrocycle backbone by thioamidation increased the interaction with the plasma membrane, enhanced binding to plasma proteins, and showed higher proteolytic stability. Thus, despite the comparable in vitro potency of the somatostatin analog **13** and its thioamidated analogs **13b, 13d**, and **13e**; the slow diffusion of the thioamidated analogs into the systemic circulation post subcutaneous injection results in a substantially longer duration of action. A property that is typically introduced into peptides by substitution with non-canonical amino acids[53] and by conjugation with molecules that bind to serum proteins[54].

In conclusion, we present an unexplored strategy to desolvate amide bonds that could be applied to improve the in vivo pharmacokinetic properties of macrocyclic peptides. This single-atom substitution allows to derive multiple (bioactive) macrocyclic analogs with varying pharmacological properties. An O to S substitution at the solvent-exposed carbonyl group did not alter the backbone conformation of the macrocyclic peptide. However, it introduces conformational restriction, a feature that is in stark contrast to N-methylation, which introduces conformational flexibility by lowering the cis/trans rotational barrier. Furthermore, the chemical stability of the thioamide linkage in the gastric and intestinal fluids, coupled with the lack of acute toxicity (Supplementary Fig. 26), warrants its use in peptide therapeutics. Although not explicitly addressed in this work, we believe that thioamide modification could be coupled with N-methylation to have better control over the bioactive conformation and efficiently desolvate the polypeptide backbone to obtain membrane-permeable and metabolically stable macrocyclic peptides. We also envision multiple thioamidation as the way forward to improve the in vivo PK of polar bioactive peptides.

## Methods

### Peptide synthesis and purification
All the peptides were synthesized following standard solid-phase Fmoc-based chemistry on 2-chlorotrityl chloride resin. The peptides were cleaved off from the resin with a mild treatment of acetic acid / 2,2,2-trifluoroethane (TFE) solution in dichloromethane (DCM) (3:1:6). The head-to-tail cyclization was performed with diphenylphosphorylacid azide (DPPA) (3eq), applying the solid base method using NaHCO$_3$ (5 eq) in DMF at a concentration of 0.1 mM. Before purification, the protecting groups were removed in solution by treating the peptides with 50% TFA in DCM and 2% TIPS. Peptides were purified by RP-HPLC, and purity was assessed by ESI-MS and analytical HPLC. Detailed procedure for the synthesis of thioamidated building blocks[55] and on resin N-methylation[56] is provided in Supplementary Information.

### NMR experiments
Spectra were recorded on a Bruker AV 500 MHz spectrometer at 25 °C. Chemical shifts are reported in parts per million (ppm) from tetramethylsilane (TMS) ($\delta = 0$) and were measured relative to the solvent in which the sample was analyzed (DMSO-$d_6$: $\delta$ 2.49 ppm, CDCl$_3$: $\delta$ 7.26 ppm). A detailed description of the pulse programs, structure calculation methods, and HDX protocol are provided in Supplementary Information.

### Docking studies
All the coordinates of the SSTR2 in complex with octreotide were fetched from the PDB entry 7T11. The structure of SSTR2 was prepared at pH 7.4 using Protein Preparation Wizard in Schrödinger Suite (Schrödinger Release 2020-2). The coordinates of the average solution structures of the peptides (**13, 13a–f**) were used for the docking studies. The peptides were docked into the SSTR2 binding pocket using the Standard Precision (SP)-peptide and flexible ligand sampling settings available in the Glide module of Schrödinger suite. The receptor-ligand interactions were visualized in Maestro GUI (version 12.04.072). Details in Supplementary Information.

### Assessment of lipophilicity
A detailed protocol of the initial lipophilicity assessment of the peptides by HPLC retention time analysis, AlogP calculation, and partition coefficient studies are provided in Supplementary Information.

### PAMPA and Caco-2 permeability
96-well BD Corning® pre-coated PAMPA plates were used following the supplier's protocol, details in Supplementary Information. The Caco-2 cell (HTB-37™ from ATCC, procured from NCCS, Pune) monolayer was washed and pre-incubated at 37 °C and 5% CO$_2$ for 30 min with HBSS-HEPES buffer containing 1% DMSO prior to experimentation. The assay was performed 21 days post-seeding of cells. TEER values were measured before starting and after the completion of the assay. Acceptor and donor well concentrations of the peptides and controls were assessed through LC-MS/MS. Further experimental details are provided in Supplementary Information.

### Assessment of in vitro proteolytic stability
A stability assessment of the peptides was done in simulated gastric (SGF) and intestinal (SIF) fluids and human liver microsomes. The degradation kinetics of the aliquoted samples at desired time points were analyzed through RP-HPLC. Details in Supplementary Information.

### Stability and protein binding in blood plasma
Blood was freshly drawn from three healthy adults, two male and one female of 26–28 year old. All participants gave written informed consent before contributing blood samples. The plasma was separated by centrifugation. The study was reviewed and approved by the Institutional Human Ethics Committee (IHEC) of the Indian Institute of Science, Bangalore. (Approval number: IHEC: 4-14032018). For checking metabolic stability, the required volume of plasma was pre-incubated at 37 °C prior to the experiment for 2 min, followed by spiking with the peptides. At different time points, plasma was aliquoted. The supernatant was separated and used for RP-HPLC quantification. Rapid Equilibrium Dialysis (RED) assay was performed to assess the plasma protein binding of the peptides following the supplier's protocol. Experimental details in Supplementary Information.

### GloSensor Assay
HEK-293T cells (CRL-3216 from ATCC, procured from NCCS, Pune) were co-transfected with human somatostatin receptor subtypes 2 & 5, plasmid cDNA, and cAMP biosensor (pGloSensorTM-22F plasmid; Promega Corp.) using PEImax. Ligand incubation was followed by the addition and incubation of forskolin (FSK) to the plate. Dose-dependent inhibition of forskolin-induced cAMP production by the peptide ligands was measured using a luminescence plate reader. An elaborate description of the assay setup has been provided in Supplementary Information.

## In vivo pharmacokinetics in rats

Animal care and in vivo procedures were conducted according to guidelines and protocols from the IEAC and CPCSEA. The work plans were reviewed and approved by the Institutional animal ethics committee of Anthem Biosciences Pvt. Ltd. and the Indian Institute of Science, Institute Animal Ethical Committee (IAEC). (Approval number: IAEC: CAF/Ethics/678/2019). All animals were housed individually in the animal facility at Anthem Biosciences Pvt. Ltd and were acclimatized for 3 days. All animals were fasted overnight before dosing, whereas access to water was provided ad libitum. Peptides were administered orally and via s.c. route and blood samples were collected at different time points through a jugular vein catheter. Dosing, sample collection, quantification, and other details as described in Supplementary Information.

## In vivo efficacy and toxicity in rats

Animal housing and acclimatization were done at Central Animal Facility, IISc Bangalore, as mentioned in the previous section, following protocols from the IEAC and CPCSEA. Peptides were administered subcutaneously, and blood samples were collected at different time points from the orbital sinus for efficacy studies. Samples were processed as discussed in Supplementary Information, and sandwich ELISA was performed to measure the growth hormone level in blood plasma. For acute toxicity, after peptide injection (s.c.), blood samples were collected on the 14th day for liver and kidney function tests. Dosing and other details as described in Supplementary Information.

## Statistical analysis

All the data in this manuscript are described as mean ± SEM unless otherwise stated. The $P$ values for statistical significance were analyzed by a one-tailed unpaired $t$-test using the GraphPad Prism software (v.8.0.1). $*P < 0.05$, $**P < 0.01$, $***P < 0.001$, $****P < 0.0001$, ns (non-significant) $> 0.05$.

## Reporting summary

Further information on research design is available in the Nature Portfolio Reporting Summary linked to this article.

## Data availability

All other data are available in the supplementary information files. CryoEM structure of somatostatin receptor 2 in complex with Octreotide (7T11) used for the docking study is available in the Protein Data Bank. Source data are provided with this paper.

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

## Acknowledgements

We acknowledge Sunita Prakash at the MS facility, and staff of the NMR facility, IISc for support. We are grateful to Prof. R. Sowdhamini for her generous support in providing access to the Schrödinger suite of programs. We thank Prof. Kartik Sunagar and Muralidhar Nayak for access to the Triple Quadrupole LC-MS/MS. We thank Dr. Ritwik Kavathekar from Biovia Dassault Systemes for his continued support with Discovery Studio. J.C. acknowledge DST-SERB (CRG/2021/000753) and DBT-IISc partnership program for funding the research. We thank DST-FIST, UGC-CAS, DBT-IISc partnership program, and MHRD for infrastructural support. P.G. thanks MHRD for research fellowship. N.R. acknowledges DBT for research fellowships (J.R.F. and S.R.F.). M.P. acknowledges AcSIR and CSIR for research fellowships (J.R.F. and S.R.F.).

## Author contributions

P.G., N.R., H.V., and J.C. designed research. P.G., N.R., H.V., M.P., S.C., B.K., C.M.D., A.S.R., D.M.B., and G.J. performed experiments and collected data. P.G., N.R., and H.V. designed and synthesized the peptides, performed in vitro PK experiments. P.G. acquired NMR and performed NMR based studies. M.P. and P.N.Y. contributed to the in vitro SSTR studies. S.C. performed the docking. G.J. performed the Caco-2 studies. C.M.D., A.S.R., and S.S. performed the in vivo PK studies. P.G. and D.M.B. performed the in vivo efficacy studies. P.G. and J.C. wrote the paper with inputs from all the authors.

## Competing interests

The authors declare no competing interests.
