## [Peer review file · Nature Communications]

REVIEWER COMMENTS

Reviewer #1 (Remarks to the Author):

The research paper by Ghosh and colleagues reports on the effect of masking the amide hydrogen bond acceptor ($>C=O$), through a thioamide substitution ($>C=S$), on passive membrane permeability of peptides and potentially enhancing their oral bioavailability. The low bioavailability of peptides following oral administration represents a major challenge and therefore, new approaches that enhance the oral bioavailability of peptide would represent a major advance in the field of peptide therapeutics. Unfortunately, in my opinion, this paper is not suitable to be published in Nature Communication, especially for relevant limitations in experimental design and analysis.

Currently, N-methylation is the most dominant chemical modification to improve the oral bioavailability of cyclic peptides. N-methylation facilitates the passive permeability of peptides across the intestinal membrane by lowering the desolvation penalty of NH group in the amide bonds and providing protection against plasma proteases. The authors failed to compare the oral or subcutaneous bioavailability of their thioamidated compounds to reported N-methylated counterparts.

The concept of increasing the lipophilicity of peptides by thioamidation is not new. Artis et al. (J. Am. Chem. Soc. 1998, 120, 47, 12200–12206) reported that substituting a single atom, O to S (amide to thioamide), in a peptide bond resulted in decreased hydrogen bonding-accepting ability of the thioamide moiety. The effect of thioamide substitution on the enhancing the stability of peptide is not new. Chen et al. (J. Am. Chem. Soc. 2017, 139, 46, 16688–16695) reported that a single thioamide modification of the GLP-1 peptide provided the GLP-1 thiopeptide with higher proteolytic plasma stability. The GLP-1 thiopeptide also maintained in vivo activity in rats after intraperitoneal injection.

Professor Chatterjee group (Khatri et al. Angew. Chem. Int. Ed. 2021, 60, 24870 –24874) reported that substituting the oxygen atom of the amide bond within a polypeptide backbone with sulfur atom desolvates the thioamide bond. They showed that this modification increases lipophilicity of dipeptides. They also demonstrated that thioamide substitution can increase protein stability. Utilizing the thioamide substitution to enhance oral bioavailability would represent a significant advance in the field of peptide and protein therapeutics. However, this reviewer is not convinced that Gosh et al. provided convincing data that they accomplished this aim.

The main concerns regarding the data analysis and other general comments are listed below:

- Ghosh et al. synthesized the monothionated regioisomers of compound 10 (Figure 1C) of a well-studied macrocycle cyclo(-D-Leu-Leu-Leu-D-Pro-Tyr-Leu-) (Rezai et al. J. AM. CHEM. SOC. 2006, 128, 2510-2511). Rezai et al. evaluated the effect of conformation on membrane permeability of nine cyclic hexapeptide diastereomers based on the sequence cyclo[Leu-Leu-Leu-Leu-Pro-Tyr]. compound 9 in Rezai et al. (i.e., Compound 10 in Ghosh et al.) displayed the lowest membrane permeability. Rezai et al. have identified a cyclic peptide analog of 9 without modifications (i.e., no N-methylation) which displayed passive membrane diffusion rate parallels that of cyclosporine A (2 order of magnitude better than parent compound 9). In their studies, Rezai et al. highlighted effect of H-bonding and solvation of NH groups on membrane permeability. It is not clear why the authors did not compare the reported N-Methylated analogs of compound 10 (i.e., compound 1 in Rezai et al.) in their experiments in Figures 1 and 3?

- It is not clear why the authors chose the small molecule propranolol as control vs. cyclosporin A or compound 1 as in Rezai et al.

- The bidirectional Caco-2 permeability (Figure 1H) data is not convincing. The authors showed that that the parent compound 10 has an efflux ratio (B-A Papp to A-B Papp) of 16X (efflux ratio $>2X$ represents poor oral bioavailability). Why did the authors not report the B-A ratio without the P-gp

efflux pump inhibitor Elacridar? Moreover, the data with Elacridar is not relevant for the in vivo PK study.

- Figure 3D: The C_{max} for all compounds are at low concentration 30-40 ng/ml (4-fold relative to parent peptide of 10 nM). The use of 230-fold ratio of AUC is not appropriate given that the higher AUC is due to higher circulation half-life through higher binding to plasma proteins as shown in Figure 3C. The word "concentration" in "The stark improvement in the plasma concentration" should be replaced by "exposure" since we are referring to AUC. Again, no data to compare their best molecules with reported N-methylated peptides to show advantage/similar performance over current methods.

- It is hard to compare the PK data to other reported peptides given that the authors did not show the actual oral bioavailability F, which can be obtained only by performing the PK experiments using the IV dosing in addition to oral dosing. The lack of solid data on oral bioavailability is a major concern for this paper.

- The authors stated that thioamidation failed to improve the intestinal permeability of the known polar macrocycle 11. They then focused on subcutaneous (s.c.) injection which is a common route for most clinically used polar peptides. There are a number of concerns about the s.c. study. The half-life for compound 11 in human plasma (~15 min, Figure S18) was slightly improved but still short (40 min for best analog). For the s.c. PK study, is the dose 250 mg/kg as stated in Figure 4C legend or 250 µg/kg as stated in Text and Supporting information? Why did the authors not report the AUC values 11 and its analogs as they stated for the oral dosing of compound 10?

- Finally, and most importantly, Biron et al. (Angew. Chem. Int. Ed. 2008, 47, 2595 –2599) reported that that multiple N-methylation of same cyclic somatostatin peptide analog 11 improved its oral bioavailability without modifying its biological activity and selectivity. Biron et al. performed the PK studies on their best cyclic peptide in rats (10 mg/kg oral and 1 mg/kg IV) and obtained a very good oral bioavailability (F = 9.9%) which is higher than that reported for clinically used and orally bioavailable somatostatin analog, octreotide (F ~1%).

Therefore, it was difficult for this reviewer to evaluate the potential of the thioamidation technology since the proper controls (e.g., octreotide, reported N-methylated compound 11) are lacking.

Reviewer #2 (Remarks to the Author):

The manuscript by Ghosh, et al. reports systematic studies of the how replacement of the C=O group of an amide bond with a thioamide influences the properties of cyclic peptides. It is found that lipophilicity is increased, which results in a lower aqueous solubility and a higher cell permeability. Thioamidation was also found to increase metabolic stability. Taken together, these effects translate into major increases in bioavailability on oral administration. As exemplified with a somatostatin analogue thioamidation may also be compatible with a retained pharmacodynamic effect. This manuscript reports a promising, novel approach to improve the cell permeability and oral bioavailability of cyclic peptides. With only a few exceptions the conclusions are properly supported by the experimental data. In addition, the manuscript is well organized and well written. I recommend that it is published after having been revised as listed below.

Major

1) Most chemical structures and some of the text in the figures are too small to read. For instance, the structure of 10 in Figure 1C can hardly be interpreted and the connection to the table via boldface a-f is impossible to read. In addition, yellow text, line or bars in figures can not be read. Figures 1, 2 and 4 need to be adjusted to show the results more clearly.

2) The discussion of the differences in permeability of cyclic peptides 10 and 10a-f on page 7 is an

important part of the manuscript, but written so that it is somewhat difficult to follow the line of thought. Here are some reasons that I struggled with this part: a) the generic structure in Figure 2A is too small and color coding of the NH groups is too faint; b) the half-lives for the protons in 10c are not significantly slower than in 10 (in contrast to what is written on page 7); c) lines 171-174 present a conclusion from the HDX study – this conclusion would be better placed after the results from the HDX exchange in CDCl₃ have been presented.

3) The NMR based conformational analysis discussed on page 8 is a) based on data that is prone to be influenced by experimental error and b) has been conducted using a method that drives the analysis towards one virtual (unreal) conformation, or a set of similar virtual conformations.

a) Interatomic distances cannot be accurately determined from a single NOESY or ROESY spectrum with an arbitrary mixing time. Instead NOESY buildups with several different mixing times should be acquired, then the linear part of the initial build-ups can be used to determine distances. Use of ROESY buildups is cumbersome as these need correction for offset effects and may suffer from TOCSY-type artifacts for strongly coupled proton pairs.

b) With very few exceptions NMR spectra acquired at room temperature constitute a time-average from multiple rapidly equilibrating conformations. Consequently, they need to be deconvoluted into the contributions from the individual conformations and not interpreted as a single, virtual (average) conformation that satisfies all NOE restraints (Cicero, et al., *J. Org. Chem.* 1999, 64, 11, 3979). Use of NMR derived restraints in MD simulations, as done in the manuscript, will provide such a virtual conformation. Deconvolution of the NMR data into real conformations can be done using Mestrenova's Stereofitter or the NAMFIS algorithm (Cicero, et. al., *J. Am. Chem. Soc.* 1995, 117, 1027).

Either the NMR based conformational analysis should be redone using a state-of-the-art method or removed. An unrestrained MD-simulation of each cyclic peptide might be just as informative.

4) Figures S6-S10, S14-S16 and S19-S24 are missing in the Supporting Information.

5) The chromatograms of cyclic peptides 10c, 10d and 11f contain large impurities close to the main peaks (5-10%?). Presumably the impurities originate from closely related peptides. I am not too worried that this will have influenced the conclusions in the manuscript, but it will have to be an editorial decision whether or not to accept the impurities.

Minor

1) Error bars are given in most figures. The number of repeats should be given in the legend to further inform the reader about the quality of the experimental data.

2) Page 5, lines 126-128 and Figure 1A. I do not agree that HBA masking has a greater impact on lipophilicity than HBD masking for 1–7. This is only the case for compound 2, whereas differences between HBA and HBD masking are not significant for 1, 3, 4 and 7. For 5 and 6 differences are barely significant, but go either way. My conclusion is that HBA and HBD masking in this dipeptide model have an equal effect on lipophilicity.

3) Page 6, line 142. The solubility of 10 and 10a-f is mentioned, but no reference is given to Figure S3A which shows the data.

4) Figure 1H. Caco2 cell permeability has only been determined for 10a, 10e and 10f which have high PAMPA permeabilities. I recommend that the Caco2 permeability is also determined for one of the three cyclic peptides that have low PAMPA permeability, e.g. 10d, to investigate the correlation between PAMPA and Caco2 permeabilities better.

5) The MD simulations illustrated in Figure 2D have only been conducted for 2 ns. This appears to be a very short time for a cyclic peptide for which rotation about amide bonds is restricted.

6) Page 9, lines 219-222. a) I assume that line 220 should read "hypothesis that reduced desolvation of the...". b) Similarly, I assume that line 221 should read "through increased solvent shielding of the..".

7) Page 17, lines 429-431. One gets the impression that N-methylation of an amide bond introduces conformational restriction when reading this sentence, but the opposite is true, i.e. that N-methylation results in increased conformational flexibility as both rotamers about the amide bond become populated.

Reviewer #3 (Remarks to the Author):

This paper by Ghosh et al describes the substitution of oxygen by sulphur in peptide bonds, in order to improve the pharmacokinetic properties of (macrocyclic) peptides. The results are clearly described and the overall message (improved proteolytic stability and membrane passage) is generally well-founded. Even though the data are promising, the paper is still premature and additional experiments are needed to improve the manuscript. More specifically, the genericity of the approach is not demonstrated and taken together, the title is not supported enough as only one particular type of cyclic peptide has been investigated, namely one type of cyclic hexapeptide (type 9/10/11 in manuscript). The diversity of tests presented is very relevant, but some techniques are not elaborated enough. For example, the conformational impact of the thioamide insertion is not clear. The conformational analysis is poorly described and figures in the ESI even lack. Hence, the conformational impact of the thioamide inclusion is not detailed enough to understand what this chemical modification does to the overall and/or local conformation. It is therefore strongly advised to improve these sections.

Additionally, it is not described how this specific modification influences the proteolytic stability, since it is not mentioned 'where' the protection takes place, meaning which amide/peptide bonds are positively impacted by this modification. Is it only the thioamide bond itself, or the adjacent ones as well? No metabolites are provided, neither for the SGF, not for the SIF or plasma stability assays. Some parts of the manuscript can also not be judged as parts of the ESI are missing.

The application on other types of linear and cyclic peptides is advised in order to prove the more generic advantage structurally diverse peptides may have through this change in the peptide backbone.

Minor comments are:

- sentence on lines 4-6 of page 6 needs revision, as it makes no sense
- inclusion or reference to the synthetic protocols of the thioamidated sequences
- page 5: cyclopenta- or cyclohexa- (not cyclicpenta...)
- page 5: last line when referring to (Fig 1C and 1D), S2 and S3A lack. The HPLC gradient should also be stated in the caption
- page 6: the B-A without Elacridar experiment is missing in the graph for the different analogues
- page 7: not DMSO-d6 and CDCl3 were used, but mixes with D2O and MeOD-d4. This should be corrected in the text and the authors should explain why these mixes were used.
- in figure 2D and in the main text, it should be mentioned which exact thioamide insertions conferred conformational stability
- R2 value of 0.68 can hardly be named 'good'
- several SXX figures lack in ESI
- in many graphs the statistical significance lacks (this needs serious attention)
- page 10: the conclusion '...indicating the cleavage of every amide bond.' is not supported as no metabolic analysis is presented. The cleavage can occur at 1 or 2 spots only.
- the error bars in Fig3D are too high, making the values and 'fold improvements' uncertain.
- caption Fig4 should be revised. mg should be replaced by microgram, i guess. It is also stunning that SRIF has an effect of 24h, while it has a t1/2 of 3 minutes. How is this possible?

REVIEWER COMMENTS AND POINT-WISE RESPONSE

Reviewer #1 (Remarks to the Author):

- The research paper by Ghosh and colleagues reports on the effect of masking the amide hydrogen bond acceptor ($>C=O$), through a thioamide substitution ($>C=S$), on passive membrane permeability of peptides and potentially enhancing their oral bioavailability. The low bioavailability of peptides following oral administration represents a major challenge and therefore, new approaches that enhance the oral bioavailability of peptide would represent a major advance in the field of peptide therapeutics. Unfortunately, in my opinion, this paper is not suitable to be published in Nature Communication, especially for relevant limitations in experimental design and analysis.

We wish to thank the reviewer for the critical assessment of our manuscript and the brilliant suggestions that have enormously helped us to demonstrate the strength of the thioamidation to improve the permeability and bioavailability of macrocyclic peptides. Nevertheless, we agree with the reviewer that our first submission was limited regarding experimental design and analysis. This has been thoroughly addressed in the current version.

- Currently, N-methylation is the most dominant chemical modification to improve the oral bioavailability of cyclic peptides. N-methylation facilitates the passive permeability of peptides across the intestinal membrane by lowering the desolvation penalty of NH group in the amide bonds and providing protection against plasma proteases. The authors failed to compare the oral or subcutaneous bioavailability of their thioamidated compounds to reported N-methylated counterparts.

This was a precious suggestion. We have now compared the passive permeability, metabolic stability (gastrointestinal fluids, human liver microsomes, human plasma), and plasma protein binding of the thioamidated hexapeptides with their N-methylated counterparts. Based on the in vitro pharmacokinetic properties, we selected two membrane-permeable and metabolically stable N-methylated peptides and performed their in vivo PK study through oral and intravenous administration in rats. The in vitro and in vivo PK study results strongly support the comparable performance of thioamidation and N-methylation in improving the PK properties of cyclic peptides.

The current **Fig. 4** in the manuscript along with the accompanying text on pages 14-17 depicts the comparison of the thioamidated vs. N-methylated analogs of **10**.

Fig.4. 1A-6A refer to the amide bonds in **10**. The yellow and the blue bars in the individual panels represent the thioamidated (**10a-10f**) and the N-methylated (**10g-10k**) counterparts, respectively, at a specific amide bond. (A) Octanol-water partition coefficient ($\log D_{7.4}$). (B) Membrane permeability (P_e) of the macrocyclic peptides determined by the PAMPA. (C) Extent of the peptides to bind non-specifically to plasma proteins determined through Rapid Equilibrium Dialysis (RED) represented in percentage (%). Half-lives ($t_{1/2}$) of the peptides in (D) simulated gastric fluid (SGF), and (E) simulated intestinal fluid (SIF). (F) Intrinsic clearance (CL_{int}) of the peptides in human liver microsomes deduced from their $t_{1/2}$ values. Cyclosporin A (CsA) was taken as a control (CL_{int} : $40 \pm 5.5 \mu\text{L min}^{-1} \text{mg}^{-1}$). $n = 3 \pm \text{SEM}$. Statistical significance was measured against the (all-amide) parent molecule by a one-tailed unpaired t-test. * $P < 0.05$, ** $P < 0.01$, *** $P < 0.001$, **** $P < 0.0001$, ns (non-significant) > 0.05 .

Comparison of thioamidation and N-methylation. To evaluate the performance of thioamidation in improving the permeability of macrocyclic peptides against the most dominant modification, N-methylation, we synthesized the N-methylated analogs of **10** (**10g-10k**) (Table S1). N-methylation enhanced the lipophilicity of **10**, as observed from the increased C18 retention time, reduced solubility, and increased $\log D_{7.4}$ of **10g-10k** (Fig. 4A & S13A-13B). Except for the amide bond 1A, thioamidation resulted in a more pronounced effect than N-methylation at all other sites. Subsequently, the N-

methylated analogs showed comparable passive permeability to their thioamidated counterparts, except at amide bond 6A, where the thioamidated analog showed better permeability (Fig. 4B). Curiously, either thioamidation or N-methylation of amide bond 1A followed by 5A resulted in the most permeable analogs within the series.

Macrocyclic peptides are endowed with proteolytic stability associated with their restricted conformational freedom; however, we were keen to evaluate their stability under the harsh proteolytic condition in the gastrointestinal tract⁴¹. Despite the presence of two D-amino acid residues (D-Leu¹ and D-Pro⁴) that are usually considered to confer proteolytic resistance to peptides, **10** underwent rapid degradation in Simulated Gastric Fluid (SGF) ($t_{1/2}$ 12 min), consisting of only pepsin at pH 1.2⁴² (Fig. 4D & S14). Chemical modification of an amide bond targeted by a protease increases its half-life^{43,44}. Thus, a significant increase in the half-life of the peptides by both thioamidation and N-methylation at the amide bonds 1A and 5A indicate that 1A and 5A are the direct cleavage sites of pepsin on the macrocycle, with 5A (flanked by Tyr⁵ and Leu⁶) being the primary cleavage site. Curiously, the increased half-life on thioamidation but not N-methylation at 6A suggests that it is not the direct cleavage site of pepsin. However, thioamidation at 6A perhaps slows down the cleavage at the preceding amide bond 5A and the following amide bond 1A, as observed earlier for linear peptides treated against serine proteases⁴³.

We next assessed the stability of the macrocyclic peptides in Simulated Intestinal Fluid (SIF)⁴², comprising of pancreatin, which is a mixture of several proteases produced by the exocrine cells of the porcine pancreas. While the parent peptide **10** showed low proteolytic stability ($t_{1/2}$ 16 min), the maximum protection was achieved by thioamidation at D-Leu¹ (**10a**) ($t_{1/2}$ 147 min), Tyr⁵ (**10e**) ($t_{1/2}$ 241 min), and Leu⁶ (**10f**) ($t_{1/2}$ 126 min) (Fig. 4E & S15), as observed in SGF. In addition to 1A and 5A, either

chemical modification at 2A protects the peptide from rapid degradation, highlighting the direct cleavage sites in SIF. In contrast, the increase in half-life only by thioamidation at 4A and 6A points at the indirect protection offered by thionating the preceding and following residues of 5A. Nonetheless, the protection offered by thioamidation outweighs the effect of N-methylation. The combined results of SGF and SIF highlight the benefit of thioamidating hydrophobic amino acids to protect macrocyclic peptides against proteolysis.

Human liver is the primary site for drug metabolism, decreasing the amount of drug entering the systemic circulation when administered through the oral route. Thus, we assessed the in vitro stability of the macrocyclic peptides against the drug-metabolizing cytochrome P450 (CYP) enzymes present in the human liver microsomes to determine the clearance by the liver (Fig. S16)¹⁴. The parent macrocycle, **10**, underwent fast degradation resulting in its rapid clearance (CL_{int} 225 μ L/min/mg), whereas, both thioamidation and N-methylation prevented the macrocycle from degradation and lowered its clearance with varying efficacy (Fig. 4F). We finally assessed the ex vivo stability of the macrocycles in human blood plasma (Fig. S17), which showed two folds or more enhanced half-life of **10a-f** compared to **10** ($t_{1/2}$ 8 h). The N-methylated analogs also showed comparable half-lives to the thioamidated analogs except **10j** and **10k** (Fig. S13D). This led us to assess their binding to plasma proteins (Fig. 4C), since plasma protein binding is beneficial for extending the circulation half-life of peptides. The thioamidated analogs **10c-10f** along with the N-methylated analog **10j** showed significantly higher plasma protein binding than **10**. This result indicates that compound lipophilicity induced by thioamidation results in amplified plasma protein binding.

The results from the *in vivo* studies are included in **Fig. 5** and **Table 1** along with the accompanying text on **pages 17-18**.

Fig.5. (A) Plasma concentration vs time plot of **10**, **10a**, **10e**, and **10f** after a single dose oral administration (10 mg/Kg) in male Wistar rats. $n = 3 \pm \text{SEM}$. (B) Comparison of the plasma concentration with respect to time of the macrocycles obtained by thioamidation and N-methylation of the amide bonds 1A (**10a**, **10h**) and 5A (**10e**, **10k**), following a single dose oral administration (10 mg/Kg) in male Wistar rats. $n = 3 \pm \text{SEM}$.

In vivo pharmacokinetic (PK) assessment of the thio-peptides. The improved *in vitro* pharmacokinetic properties of the thioamidated analogs led us to evaluate the absorption of the macrocyclic peptides into the systemic circulation post oral administration in male Wistar rats. **10a**, **10e**, and **10f**, which showed enhanced permeability (Fig. 4B) and improved metabolic stability than the parent macrocycle **10** (Fig. 4D-4F), were chosen for the evaluation (Fig. 5A). In addition, we also selected **10h** and **10k** to compare the effect of N-methylation versus thioamidation at the amide bonds 1A and 5A (Fig. 5B). Since we sought to evaluate the sole effect of the chemical modification on the *in vivo* PK properties of the macrocyclic peptides, we conducted the *i.v.* (intravenous) and *p.o.* (per oral) studies without using any microemulsion formulation that are known to improve the oral bioavailability of peptides⁴⁵. The low *in*

vivo permeability and metabolic stability of **10** was evident from its maximum concentration in the blood (C_{max} : 3.5 ng/ml) after a 10 mg/kg oral administration (Table 1). Furthermore, its plasma exposure rapidly declined, reaching an undetectable limit by 1 h. On the contrary, **10a**, **10e**, and **10f** achieved a C_{max} of 15 ng/ml, 47.2 ng/ml, and 61.7 ng/ml, respectively, post oral administration and could be detected till 8 h or more (Fig. 5A). The stark improvement in the plasma exposure of **10a** (26.9 h ng mL⁻¹; ~ 30 folds), **10e** (359.5 h ng mL⁻¹; ~ 400 folds), and **10f** (120.8 h ng mL⁻¹; ~ 130 folds) with respect to **10** (0.9 h ng mL⁻¹) till the time of the last measurable concentration (AUC_{last}) emphasize the contribution of this single atom substitution in improving the bioavailability of macrocyclic peptides (Table 1). Remarkably, the N-methylated analogs **10h** and **10k** show comparable C_{max} and plasma exposure (vital parameters for determining the drug exposure) to their respective thioamidated analogs post oral administration (Table 1 & S5). Although, we note a higher fraction of **10k** reaching the systemic circulation than **10e** by the oral route (represented by %F), the comparable enhancement in their plasma exposure post oral administration with respect to **10**, suggest the potential benefit of thioamidation in improving the in vivo PK of macrocyclic peptides.

Table 1. Oral and intravenous pharmacokinetic data of **10** and its selected thioamidated and N-methylated analogs in rats^a.

Table 1. In vivo pharmacokinetics in male Wistar rats											
Peptides	p.o.				Fold change	i.v.				%F	
	C_{max} (ng ml ⁻¹)	MRT _{last} (h)	AUC_{last} (ng h ml ⁻¹)			C_{max} (ng ml ⁻¹)	MRT _{last} (h)	V_{ss} (l kg ⁻¹)	CL _{obs} (ml min ⁻¹ kg ⁻¹)		AUC_{last} (ng h ml ⁻¹)
10	3.5	0.2	0.9		-	1046.7	1.6	4.1	28.4	548.1	0
10a] 1A	15.0	2.0	26.9	*	31	48.1	2.8	28.6	91.1	146.4	1.8
10h	13.1	1.3	25.9	**	29	434.6	0.4	2.3	79.5	212.2	1.2
10e] 5A	47.2	5.9	359.5	***	409	2660.9	1.5	0.7	7.5	2278.4	1.6
10k	55.9	9.0	344.9	****	393	1165.5	0.4	0.8	33.3	509.5	6.8
10f	61.7	2.1	120.8	*	138	580.0	1.1	3.2	42.2	389.7	3.1

^a $n = 3 \pm$ SEM. Statistical significance was measured by a one-tailed unpaired t-test. * $P < 0.05$, ** $P < 0.01$, *** $P < 0.001$, **** $P < 0.0001$. (%F = $(AUC_{last(p.o.)}/AUC_{last(i.v.)}) \times (100/x)$, x = p.o./i.v. dose).

- The concept of increasing the lipophilicity of peptides by thioamidation is not new. Artis et al. (J. Am. Chem. Soc. 1998, 120, 47, 12200–12206) reported that substituting a single atom, O to S (amide to thioamide), in a peptide bond resulted in decreased hydrogen bonding-accepting ability of the thioamide moiety.

We wish to highlight that Artis et al., through ab-initio calculations, studied the conformational impact of thioamidated dipeptides. While their study reports the decreased hydrogen bond accepting ability of C=S, some recent and old studies experimentally demonstrate that the acceptor capacity of the thioamide sulfur may equal or exceed that of the amide C=O group.

a. Lampkin BJ, VanVeller B. Hydrogen bond and geometry effects of thioamide backbone modifications. J. Org. Chem. 2021, 86, 18287-18291.

b. Mundlapati VR, Gautam S, Sahoo DK, Ghosh A, Biswal HS. Thioamide, a hydrogen bond acceptor in proteins and nucleic acids. J. Phys. Chem. Lett. 2017, 8, 4573-4579.

c. Hollosi M, Zewdu M, Kollat E, Majer Z, Kajtar M, Batta G, Kover K, Sandor P. Intramolecular H-bonds and thioamide rotational isomerism in thiopeptides. Int. J. Peptide Protein Res. 1990, 36, 173-181.

Moreover, it is well known that the thioamide NH is a stronger H-bond donor than the corresponding amide counterpart.

a. Alemán C. On the ability of modified peptide links to form hydrogen bonds J. Phys. Chem. A 2001, 105, 6717-6723.

b. Lee HJ, Choi YS, Lee KB, Park J, Yoon CJ. Hydrogen bonding abilities of thioamide. J. Phys. Chem. A 2002, 106, 7010-7017.

c. Dudek EP, Dudek GO. Proton Magnetic Resonance Spectra of Thiocarboxamides. J. Org. Chem. 1967, 32, 823-824.

Thus, with the opposing H-bond potential of C(S) and N(H) in thioamides, no study to date reports the increased lipophilicity of macrocyclic peptides on thioamidation and its consequence on the passive membrane permeability.

- The effect of thioamide substitution on the enhancing the stability of peptide is not new. Chen et al. (J. Am. Chem. Soc. 2017, 139, 46, 16688–16695) reported that a single thioamide modification of the GLP-1 peptide provided the GLP-1 thiopeptide with higher proteolytic plasma stability. The GLP-1 thiopeptide also maintained in vivo activity in rats after intraperitoneal injection.

We agree with the reviewer that the report of Chen et al. and earlier studies report the protection of linear peptides against proteolytic degradation by thioamide substitution, primarily composed of a single proteolytic cleavage site.

a. Mock WL, Chen JT, Tsang JW. Hydrolysis of a thiopeptide by cadmium carboxypeptidase A. Biochem. Biophys. Res. Commun. 1981, 102, 389-396.

b. Bartlett PA, Spear KL, Jacobson NE. A thioamide substrate of carboxypeptidase A. Biochemistry 1982, 21, 1608-1611.

However, in the present manuscript, we demonstrate the proteolytic stability conferred by thioamidation to macrocyclic peptides having multiple proteolytic cleavage sites against the harsh enzymes found in the gastrointestinal fluid, liver microsomes, and blood plasma.

Additionally, we demonstrate how increased lipophilicity leads to enhanced plasma protein binding, an essential component for extending the plasma exposure of drugs (AUC).

Through improved metabolic stability and membrane permeability, we sought to demonstrate how thioamidation improves the PK properties of macrocyclic peptides that has never been shown before.

Curiously, as per the reviewer's suggestion, when we compared the proteolytic stability conferred by thioamidation of the amide bond against the most commonly employed N-methylation, we were pleasantly surprised by the superior performance of thioamidation. Thus, we decided to present a figure (Fig. 4) in the main text with the comparative data, which would be extremely valuable for further developing peptide and protein therapeutics.

Fig.4. 1A-6A refer to the amide bonds in **10**. The yellow and the blue bars in the individual panels represent the thioamidated (**10a-10f**) and the N-methylated (**10g-10k**) counterparts, respectively, at a specific amide bond. Half-lives ($t_{1/2}$) of the peptides in (D) simulated gastric fluid (SGF), and (E) simulated intestinal fluid (SIF). (*adopted from Fig. 4 above*)

- Professor Chatterjee group (Khatri et al. *Angew. Chem. Int. Ed.* 2021, 60, 24870 –24874) reported that substituting the oxygen atom of the amide bond within a polypeptide backbone with sulfur atom desolvates the thioamide bond. They showed that this modification increases lipophilicity of dipeptides. They also demonstrated that thioamide substitution can increase protein stability. Utilizing the thioamide substitution to enhance oral bioavailability would represent a significant advance in the field of peptide and protein therapeutics. However, this reviewer is not convinced that Gosh et al. provided convincing data that they accomplished this aim.

We thank the reviewer for the critical evaluation and for highlighting the significance of the question addressed in the manuscript towards the advancement of the field of peptide and protein therapeutics. Following the reviewer's suggestion, our new experiments substantiate the manuscript's claims.

- The main concerns regarding the data analysis and other general comments are listed below:

- Ghosh et al. synthesized the monothionated regioisomers of compound 10 (Figure 1C) of a well-studied macrocycle cyclo(-D-Leu-Leu-Leu-D-Pro-Tyr-Leu-) (Rezai et al. *J. AM. CHEM. SOC.* 2006, 128, 2510-2511). Rezai et al. evaluated the effect of conformation on membrane permeability of nine cyclic hexapeptide diastereomers based on the sequence cyclo[Leu-Leu-Leu-Leu-Pro-Tyr]. compound 9 in Rezai et al. (i.e., Compound 10 in Ghosh et al.) displayed the lowest membrane permeability. Rezai et al. have identified a cyclic peptide analog of 9

without modifications (i.e., no N-methylation) which displayed passive membrane diffusion rate parallels that of cyclosporine A (2 order of magnitude better than parent compound 9). In their studies, Rezai et al. highlighted effect of H-bonding and solvation of NH groups on membrane permeability. It is not clear why the authors did not compare the reported N-Methylated analogs of compound 10 (i.e., compound 1 in Rezai et al.) in their experiments in Figures 1 and 3?

We thank the reviewer for this valuable suggestion. We selected the cyclic hexapeptide (10) that showed the lowest membrane permeability from Rezai et. al. to evaluate the effect of thioamidation in improving the passive membrane diffusion of cyclic peptides. We have now synthesized all the possible N-methylated analogs of 10 (10g-10k) and measured their passive membrane permeability by PAMPA. The results of which are presented in Fig. 4B.

N-methylation improved the passive permeability of 10. However, when we compared the membrane permeabilities of the thioamidated and the N-methylated analogs, we noted that N-methylation or thioamidation of certain amide bonds showed comparable behavior in improving the membrane permeability of the cyclic peptide. This result suggests that these amide bond modifications (N-methylation and thioamidation) could perhaps be used interchangeably to improve the membrane permeability of cyclic peptides.

Fig.4. 1A-6A refer to the amide bonds in 10. The yellow and the blue bars in the individual panels represent the thioamidated (10a-10f) and the N-methylated (10g-10k) counterparts, respectively, at a specific amide bond. (B) Membrane permeability (P_e) of the macrocyclic peptides determined by the PAMPA. (adopted from Fig. 4 above)

- It is not clear why the authors chose the small molecule propranolol as control vs. cyclosporin A or compound 1 as in Rezai et al.

The small molecule propranolol has >90% intestinal absorption (Artursson P, Karlsson J. Correlation between oral drug absorption in humans and apparent drug permeability coefficients in human intestinal epithelial (Caco-2) cells. Biochem. Biophys. Res. Commun. 1991, 175, 880-885), shows high permeability in PAMPA (Kansy M, Senner F, Gubernator K. Physicochemical high throughput screening: parallel artificial membrane permeation assay in the description of passive absorption processes. J. Med. Chem. 1998, 41, 1007-1010), and has been shown to be completely absorbed in humans (Chiou WL, Jeong HY, Chung SM, Wu TC. Evaluation of using dog as an animal model to study the fraction of oral dose absorbed of 43 drugs in humans. Pharm Res. 2000, 17, 135-40).

This is why it is used as a control in membrane permeability assays. Nevertheless, we have now included the highly permeable compound 1 (cyclo(-D-Leu-D-Leu-Leu-D-Leu-Pro-Tyr-) reported in Rezai et al. in our PAMPA assay.

The results are included in **Fig. 1** and the accompanying text on **pages 7-8**.

Fig.1.(C) AlogP of cyclo(-D-Leu¹-Leu-Leu-D-Pro-Tyr-Leu⁶-)(**10**), cyclo(-Ile¹-Ala-Ala-Phe-Pro-Ile-Pro⁷-)(**11**), and cyclo(-D-Leu¹-Leu-D-Pro-D-Leu-Leu-D-Ala-Pro-Leu⁸-)(**12**). The small letters denote the site of thioamidation in the individual scaffolds. (D) logD_{7.4} of **10**, **11**, **12** and their respective thioamidated analogs. (E) Membrane permeability of the macrocyclic peptides determined by the PAMPA (P_e). Propranolol and **PC** (cyclo(-D-Leu¹-Leu-D-Leu-Pro-Tyr-D-Leu⁶-)³¹) were used as the markers for transcellular transport. (*adopted from Fig. 1*)

We next sought to investigate the origin of permeability on thioamidation of **10**, a scaffold that has been extensively investigated. As noted in the previous reports, **10** showed a moderate permeability (2.45×10^{-6} cm/s), which significantly improved on thioamidation at D-Leu¹ (**10a**), Tyr⁵ (**10e**), and Leu⁶ (**10f**) with marginal improvement at the other sites (**10b-10d**) (Fig. 1E). What was striking is the marked improvement in permeability of **10a** (10.75×10^{-6} cm/s) by this single atom substitution against **PC** (15.76×10^{-6} cm/s), derived by altering the amino acid chirality at 3 sites (Fig. 1E).

- The bidirectional Caco-2 permeability (Figure 1H) data is not convincing. The authors showed that the parent compound **10** has an efflux ratio (B-A P_{app} to A-B P_{app}) of 16X (efflux ratio >2X represents poor oral bioavailability). Why did the authors not report the B-A ratio without the P-gp efflux pump inhibitor Elacridar? Moreover, the data with Elacridar is not relevant for the in vivo PK study.

We agree with the reviewer that the bidirectional Caco-2 permeability data was incomplete. We have included the A to B and B to A permeabilities for all the cyclic peptides. We note that all the compounds show a high efflux ratio, as shown by the parent compound. However, when the efflux pump was blocked, we saw improved permeability of the cyclic peptides. The results are now reported in Fig. 3.

*Furthermore, we observe improved correlation between the PAMPA and Caco-2 permeabilities on adding the P-gp efflux pump inhibitor Elacridar. This experiment highlights the role of thioamidation in improving the passive permeability of cyclic peptides and reveals that thioamidation can improve both membrane and cell permeability, as shown in the following report: Furukawa A, Townsend CE, Schwochert J, Pye CR, Bednarek MA, Lokey RS. Passive membrane permeability in cyclic peptomer scaffolds is robust to extensive variation in side chain functionality and backbone geometry. *J. Med. Chem.* **2016**, *59*, 9503-9512.*

*It should also be noted that **10a**, **10e**, and **10f** showed higher permeability than the orally bioavailable Atenolol without Elacridar, which we have now used as a control in the experiment.*

The results are included in Figs. 3C,D and the accompanying text on pages 13-14.

Fig.3. (C) Bidirectional Caco-2 permeability in the presence and absence of P-gp efflux pump inhibitor elacridar, represented in bars as P_{app} . Propranolol and Atenolol were used markers of high and low permeability, respectively. Digoxin was used as the control substrate of P-gp efflux pumps. $n = 2 \pm$ SEM. (D) Linear correlation plot of P_e (PAMPA) vs. (AB + Elacridar) P_{app} (Caco-2). Statistical significance was measured against the (all-amide) parent molecule by a one-tailed unpaired t-test. * $P < 0.05$, ** $P < 0.01$, *** $P < 0.001$, **** $P < 0.0001$, ns (non-significant) > 0.05 . (adopted from Fig. 3)

We next determined the permeability of the peptides across confluent monolayers of human adenocarcinoma cells (Caco-2) as a model of the intestinal epithelium. **10** showed low permeability (0.76×10^{-6} cm/s) in the apical to basolateral (A-B) direction and high permeability (12.4×10^{-6} cm/s) in the basolateral to apical (B-A) direction, suggesting the involvement of efflux pumps in lowering the permeability of **10** (Fig.

3C). **10a**, **10e**, and **10f** also showed high efflux ratio (Table S4), however, they displayed enhanced permeability than **10** except **10c**, which showed rapid exchange of amide protons (Fig. 2B). When the efflux pump was inhibited³⁹ (confirmed by an efflux ratio of <2), a significant improvement in the permeability of the peptides was observed that correlated with the values obtained from the PAMPA (Fig. 3D)⁴⁰. These results suggest that thioamide substitution improves the passive permeability of cyclic peptides across the lipid bilayer in live cells. It is also worth noting that **10a**, **10e**, and **10f** show higher permeability than the orally bioavailable atenolol in the absence of efflux pump inhibitor (Fig. 3C).

- Figure 3D: The C_{max} for all compounds are at low concentration 30-40 ng/ml (4-fold relative to parent peptide of 10 nM). The use of 230-fold ratio of AUC is not appropriate given that the higher ACU is due to higher circulation half-life through higher binding to plasma proteins as shown in Figure 3C. The word "concentration" in "The stark improvement in the plasma concentration" should be replaced by "exposure" since we are referring to AUC. Again, no data to compare their best molecules with reported N-methylated peptides to show advantage/similar performance over current methods.

We agree with the referee that concentration should be replaced by exposure. However, we think the AUC ratios of the thioamidated peptides to the parent peptide following oral administration is a crucial parameter to demonstrate the benefit of the chemical modification (thioamidation) in achieving higher plasma exposure of a parent drug, which is typically done by using self-emulsifying drug delivery systems (Balakrishnan P, Lee BJ, Oh DH, Kim JO, Hong MJ, Jee JP, Kim JA, Yoo BK, Woo JS, Yong CS, Choi HG. Enhanced oral bioavailability of dexibuprofen by a novel solid self-emulsifying drug delivery system (SEDDS). Eur. J. Pharm. Biopharm. 2009, 72, 539-455).

*Although the C_{max} of the thioamidated analogs **10a**, **10e**, and **10f** is low, their enhanced plasma exposure results from improved permeability, higher binding to plasma proteins, and increased proteolytic stability due to their amplified lipophilicity imparted by thioamidation. Demonstrating the multifaceted role of thioamidation on the macrocyclic peptide backbone. As per the reviewer's recommendation, we have now included the comparison with N-methylated analogs, which clearly demonstrates the comparable performance of N-methylation and thioamidation to improve the in vivo PK of macrocyclic peptides.*

*Please refer to **Fig. 5B** and **Table 1** above.*

We also wish to highlight that in our study for absolute comparison of the thioamidation on the peptide backbone, we have not used any formulation (SEDDS/SNEDDS, etc.) to deliver the cyclic peptides that typically results in higher C_{max} and AUC values.

- It is hard to compare the PK data to other reported peptides given that the authors did not show the actual oral bioavailability F, which can be obtained only by performing the PK

experiments using the IV dosing in addition to oral dosing. The lack of solid data on oral bioavailability is a major concern for this paper.

We agree with the referee and, thus, have repeated the animal experiments to obtain data on oral bioavailability. However, we wish to highlight that although all the thioamidated and N-methylated analogs of 10 show improved oral bioavailability, the absolute bioavailability (%F) does not accurately reflect the effect of the chemical modification (thioamidation/N-methylation) in improving the plasma exposure of the macrocyclic peptide (AUC) post oral administration, which is the most important factor in eliciting a biological response.

We observe that 10f with an oral bioavailability of 3.1% shows an AUC of 120.8 ng h ml⁻¹, which is ~1/3rd the AUC of 10e (359.5 ng h ml⁻¹) with an oral bioavailability of 1.6%. Likewise, 10a, with an oral bioavailability of 1.8%, shows ~ 13 folds lower AUC (26.9 ng h ml⁻¹) than 10e.

Table 1. Oral and intravenous pharmacokinetic data of **10** and its selected thioamidated and N-methylated analogs in rats^a.

Table 1. In vivo pharmacokinetics in male Wistar rats											
Peptides	p.o.				Fold change	i.v.				%F	
	C _{max} (ng ml ⁻¹)	MRT _{last} (h)	AUC _{last} (ng h ml ⁻¹)			C _{max} (ng ml ⁻¹)	MRT _{last} (h)	V _{ss} (l kg ⁻¹)	CL _{obs} (ml min ⁻¹ kg ⁻¹)		AUC _{last} (ng h ml ⁻¹)
10	3.5	0.2	0.9		-	1046.7	1.6	4.1	28.4	548.1	0
10a] 1A	15.0	2.0	26.9	31	48.1	2.8	28.6	91.1	146.4	1.8	
10h	13.1	1.3	25.9	29	434.6	0.4	2.3	79.5	212.2	1.2	
10e] 5A	47.2	5.9	359.5	409	2660.9	1.5	0.7	7.5	2278.4	1.6	
10k	55.9	9.0	344.9	393	1165.5	0.4	0.8	33.3	509.5	6.8	
10f	61.7	2.1	120.8	138	580.0	1.1	3.2	42.2	389.7	3.1	

^an = 3 ± SEM. Statistical significance was measured by a one-tailed unpaired t-test. *P < 0.05, **P < 0.01, ***P < 0.001, ****P < 0.0001. (%F = (AUC_{last(p.o.)}/AUC_{last(i.v.)}) × (100/x), x = p.o./i.v. dose).

Since we are proposing thioamidation to improve the plasma exposure of a parent drug post oral administration for improved systemic exposure, the C_{max} and the AUC increase with respect to the AUC parent provides a more accurate estimate.

- The authors stated that thioamidation failed to improve the intestinal permeability of the known polar macrocycle 11. They then focused on subcutaneous (s.c.) injection, which is a common route for most clinically used polar peptides. There are a number of concerns about the s.c. study. The half-life for compound 11 in human plasma (~15 min, Figure S18) was slightly improved but still short (40 min for best analog). For the s.c. PK study, is the dose 250 mg/kg as stated in Figure 4C legend or 250 µg/kg as stated in Text and Supporting information? Why did the authors not report the AUC values 11 and its analogs as they stated for the oral dosing of compound 10?

We wish to thank the reviewer for pointing out the errors in the manuscript. The half-lives of the peptides in human plasma reported in Fig. 7A are in hours. We have now clearly mentioned it in the figure legend to avoid confusion. The t_{1/2} for 13 is 15 ± 0.9 h, and for 13d, t_{1/2} is 42 ± 0.4 h. The dose used for the s.c. PK study is 250 µg/kg; we have now corrected it in the manuscript. The AUC values of 13 and its analogs are reported in Table S7. However, the rapid drop in the plasma concentration of 13 against the thioamidated analogs must be noted, which correlates with their in vivo efficacy in inhibiting the release of growth hormone.

Fig.7. (A) Half-lives ($t_{1/2}$) of **13**, **13a-13f** determined in human blood plasma. (B) % protein binding of the peptides in plasma, as measured through RED. (C) The IC₅₀ values of the peptides determined from dose-dependent inhibition of forskolin-induced cAMP production in HEK293T cells expressing SSTR2 and SSTR5. $n = 3 \pm \text{SEM}$. (D) Plasma concentration vs time plot of **13**, **13a**, **13b**, **13d**, and **13e** after a single dose (250 $\mu\text{g}/\text{Kg}$) s.c. administration in male Sprague-Dawley rats. The rapid drop in the concentration of **13** as opposed to its four thio analogs in plasma at later time points (2-8 h) are shown in the inset. (E) Inhibition of rat growth hormone (rGH) levels over 48 hours post single dose (250 $\mu\text{g}/\text{Kg}$) peptide (**13**, **13a**, **13b**, **13d**, **13e**) and PBS (Untreated) administration via the s.c. route. (F) Bar plots represent the area under the concentration-time curve (AUCs) of rGH for (F) 0-48 h, (G) 0-24 h, and (H) 24-48 h. $n = 3 \pm \text{SEM}$ (A to D); $n = 6 \pm \text{SEM}$ (E). Statistical significance was measured against the (all-amide) parent molecule by a one-tailed unpaired t-test. * $P < 0.05$, ** $P < 0.01$, *** $P < 0.001$, **** $P < 0.0001$, ns (non-significant) > 0.05 .

Table S7. Single dose subcutaneous pharmacokinetic parameters of **13**, **13a-f** in male SD rats.

PK Parameters	13	13a	13b	13d	13e
C_{max} (ng ml ⁻¹)	205.2 ± 18.4	152.2 ± 27.4	98.3 ± 26.4	101.6 ± 9.5	59.1 ± 8.6
AUC _{last} (ng h ml ⁻¹)	220.8 ± 47.6	228.5 ± 23.9	154.2 ± 41.3	146.2 ± 24.3	121.5 ± 20.2
$t_{1/2}$ (h)	0.3 ± 0.05	0.7 ± 0.1	0.6 ± 0.2	1.1 ± 0.4	0.6 ± 0.3
MRT _{last} (h)	0.7 ± 0.05	1.1 ± 0.1	1.0 ± 0.04	1.1 ± 0.2	1.5 ± 0.1

The text below from pages 20-21 describes the PK results.

Prolonged plasma exposure of thio-peptides following s.c. injection. With the failed attempt to improve the intestinal permeability of the polar macrocycle **13** by single atom substitution, we focused on subcutaneous (s.c.) injection, which is the most common route of therapeutic peptide administration in the development pipeline⁴⁸. Wherein the major challenge that needs to be addressed is their rapid clearance from plasma owing to fast enzymatic degradation and renal clearance attributing to short half-lives⁴⁹. Thus, we tested the stability of the thioamidated macrocycles **13a-13f** in plasma (Fig. 7A). As observed for the hydrophobic macrocycle **10** (Fig. S17), a single amide to thioamide substitution in most cases increased the half-life of **13** ($t_{1/2}$ 15 h) by two folds or more. Likewise, we observe a significant increase in the binding to plasma proteins by **13a-13f** (Fig. 7B), which correlates with their increased lipophilicity ($\log D_{7.4}$) (Fig. S19C). Subsequently, we selected **13a**, **13b**, **13d**, and **13e** for assessing their plasma exposure post single dose subcutaneous injection into male Sprague Dawley rats. The parent macrocycle **13** was efficiently absorbed into plasma; however, its concentration rapidly dropped within 2 h (8 ng/mL). On the contrary, **13a** (40 ng/mL), **13b** (29 ng/mL), **13d** (17 ng/mL), and **13e** (39 ng/mL) showed higher plasma concentration at 2 h and were slowly absorbed into the plasma than **13** (Fig. 7C & Table S7). Thus, the enhanced lipophilicity of the macrocyclic peptides by thioamidation result in higher binding to the plasma membrane and/or with proteins like albumin at the site of injection. This leads to their reduced diffusion rate and longer retention at the site of injection.

The text below from page 23 describes the in vivo efficacy results.

In vivo inhibition of GH. Finally, we estimated the in vivo efficacy of **13** and the thioamidated analogs **13b**, **13d**, and **13e** in suppressing the release of growth

hormone (GH) post subcutaneous injection at 250 $\mu\text{g}/\text{kg}$ dosage (Fig. 7E). **13a** was also included as a negative control to assess its effect on GH release. The rats were bled from their orbital sinus, and the plasma GH levels were evaluated through sandwich ELISA. Except **13a**, we observed a significant inhibition of unstimulated GH release by **13**, **13b**, **13d**, and **13e**. Remarkably, despite the strongest inhibition of GH release at the initial time point (1h) by **13**, the thioamidated analogs showed better inhibition of GH release at the subsequent time points throughout the course of experiment over 48 h (Fig. 7E). This correlates well with the high plasma concentration of **13** at the initial time points and a rapid drop within 2 h of s.c. administration (Fig. 7C). **13** has the least potency in inhibiting the GH release over 0-48 h (Fig. 7F), wherein it shows substantial inhibition only within 24h (Fig. 7G). In contrast, the inhibitory effect lasts longer in **13b**, **13d**, and **13e** showing a significant reduction in plasma GH levels within 24-48 h (Fig. 7H). **13d** shows the maximum inhibition of GH release, which is related to its longer retention in the plasma (Fig. 7C) due to its high plasma stability (Fig. 7A) than the other thioamidated analogs. The rapid drop in plasma concentration of **13** is responsible for its least in vivo efficacy. In comparison, the increased metabolic stability and lipophilicity of the macrocyclic peptide on thioamidation resulted in long-acting analogs that slowly diffused into plasma from the site of injection.

- Finally, and most importantly, Biron et al. (Angew. Chem. Int. Ed. 2008, 47, 2595 –2599) reported that multiple N-methylation of same cyclic somatostatin peptide analog **11** improved its oral bioavailability without modifying its biological activity and selectivity. Biron et al. performed the PK studies on their best cyclic peptide in rats (10 mg/kg oral and 1 mg/kg IV) and obtained a very good oral bioavailability (F = 9.9%) which is higher than that reported for clinically used and orally bioavailable somatostatin analog, octreotide (F ~1%).

*Thank you for highlighting this point. The present manuscript is the very first report on improving the PK properties of macrocyclic peptides by **thioamidation at a single amide bond**. Our comparative results on N-methylation and thioamidation of **10** at a single amide bond clearly highlight the potential of thioamidation in improving the PK properties of cyclic peptides. The report of Biron et al. adopted a combinatorial approach in deriving an orally*

bioavailable analog through **multiple N-methylation**. However, the article does not report the *in vivo* effect of the peptide in inhibiting the release of growth hormone. Our consistent efforts over the years have simplified the synthesis of **monothioamidated peptides**.

a. Mukherjee S, Verma H, Chatterjee J. Efficient Site-Specific Incorporation of Thioamides into Peptides on a Solid Support. *Org. Lett.* **2015**, 17, 3150-3153.

b. Mukherjee S, Chatterjee J. Suppressing the epimerization of endothioamide peptides during Fmoc/t-Bu-based solid phase peptide synthesis. *J. Pept. Sci.* **2016**, 22, 664-672.

c. Khatri B, Bhat P, Chatterjee J. Convenient synthesis of thioamidated peptides and proteins. *J. Pept. Sci.* **2020**, 26, e3248.

d. Khatri B, Raj N, Chatterjee J. Opportunities and challenges in the synthesis of thioamidated peptides. *Methods Enzymol.* **2021**, 656, 27-57.

Nevertheless, during the present work, we realized that for polar bioactive peptides, multiple thioamidation (analogous to multiple N-methylation) might produce comparable PK properties to multiple N-methylated analogs. However, unfortunately, to date, synthetic challenges preclude the synthesis of multiple thioamidated cyclic peptides in a site-selective manner. We have realized the potential of multiple thioamidation and are actively working on deriving a synthetic protocol to obtain di- and tri-thioamidated peptides.

- Therefore, it was difficult for this reviewer to evaluate the potential of the thioamidation technology since the proper controls (e.g., octreotide, reported N-methylated compound 11) are lacking.

In the present revision, we have attempted to provide additional data as per the reviewer's recommendation including the appropriate controls to justify the potential of thioamidation technology to improve the PK properties of cyclic peptides.

Reviewer #2 (Remarks to the Author):

The manuscript by Ghosh, et al. reports systematic studies of the how replacement of the C=O group of an amide bond with a thioamide influences the properties of cyclic peptides. It is found that lipophilicity is increased, which results in a lower aqueous solubility and a higher cell permeability. Thioamidation was also found to increase metabolic stability. Taken together, these effects translate into major increases in bioavailability on oral administration. As exemplified with a somatostatin analogue thioamidation may also be compatible with a retained pharmacodynamic effect.

This manuscript reports a promising, novel approach to improve the cell permeability and oral bioavailability of cyclic peptides. With only a few exceptions the conclusions are properly supported by the experimental data. In addition, the manuscript is well organized and well written. I recommend that it is published after having been revised as listed below.

We sincerely appreciate and thank the reviewer for the extremely careful and thorough assessment of our manuscript and for providing valuable suggestions for improving its readability and impact.

Major

1) Most chemical structures and some of the text in the figures are too small to read. For instance, the structure of 10 in Figure 1C can hardly be interpreted and the connection to the table via boldface a-f is impossible to read. In addition, yellow text, line or bars in figures can not be read. Figures 1, 2 and 4 need to be adjusted to show the results more clearly.

Thank you for suggesting the improvement. We have now taken extra care to label the figures and choose the color of the bars for better readability.

2) The discussion of the differences in permeability of cyclic peptides 10 and 10a-f on page 7 is an important part of the manuscript, but written so that it is somewhat difficult to follow the line of thought. Here are some reasons that I struggled with this part: a) the generic structure in Figure 2A is too small and color coding of the NH groups is too faint; b) the half-lives for the protons in 10c are not significantly slower than in 10 (in contrast to what is written on page 7); c) lines 171-174 present a conclusion from the HDX study – this conclusion would be better placed after the results from the HDX exchange in CDCl₃ have been presented.

Thank you for suggesting the improvement.

a) We have now increased the size of the generic figure and revised the color coding for easy readability.

b) We agree that the half-lives for the protons in 10c are not significantly slower than in 10, and thus the text has been revised accordingly.

*c) As per the suggestion, we have placed the conclusion of the study after the results of the HDX in CDCl₃. We have also included the HDX data for all the compounds in **Figs. 2B,C**.*

The results are included in **Figs. A-D** and the accompanying text on **pages 8-9**.

Fig.2. (A) Chemical structure of **10**, with each amide proton color coded. The small letters denote the site of thioamidation. (B) Dot plots representing the half-lives ($t_{1/2}$) of the individual amide protons of **10**, **10a-f** deduced from the HDX experiment in 20% D₂O in DMSO-*d*₆, and (C) in 20% CD₃OD in CDCl₃. The slow exchanging amide protons whose $t_{1/2}$ could not be determined have been represented as % remaining after 24 hours. (*adopted from Fig. 2*)

As hydrogen bonding and solvation of NH groups play an important role in the permeability of macrocycles, we determined the solvent accessibility of individual amide protons in **10** and **10a-10f** by hydrogen-deuterium exchange (HDX) through NMR spectroscopy (Fig. 2A-2C). Since the cyclic peptides have moderate solubility in water (Fig. S3C), we used DMSO-*d*₆:D₂O (8:2) to mimic the aqueous polar environment and CDCl₃:CD₃OD (8:2) as a membrane mimic, where D₂O and CD₃OD act as the deuterium donor, respectively. We note that thioamidation at D-Leu¹ (**10a**), Tyr⁵ (**10e**), and Leu⁶ (**10f**) results in slow exchange of the amide protons with respect to **10**, indicative of enhanced solvent shielding. In comparison, a fast exchange indicating solvent exposure of the amide protons is observed for **10b**, **10c**, and **10d** in DMSO-*d*₆ (Fig. 2B & S4). In CDCl₃, the amide protons exchange at a slower rate than in DMSO-*d*₆, suggesting their solvent-shielded nature. Most amide protons in **10a** are solvent-shielded in the nonpolar environment, followed by **10d**, **10e**, and **10f**, while **10b** and **10c** show faster exchange than **10** (Fig. 2C & S5).

These results indicate that **10a**, **10e**, and **10f** have a higher tendency than **10** to transfer from a polar to a nonpolar environment. Once in a nonpolar environment, the

adoption of a membranophilic conformation wherein the amide protons are solvent-shielded facilitates the translocation of the macrocyclic peptide. On the contrary, the high solvent exposure of amide protons in **10b**, **10c**, and **10d** as compared to **10** in DMSO- d_6 suggests a lower tendency of these three analogs to transfer into a nonpolar environment. Moreover, the high solvent exposure of **10b** and **10c** in $CDCl_3$ is suggestive of a polar conformation within a nonpolar environment, which would be disfavored. Thus, thioamide substitution increases the passive permeability of **10a**, **10e**, and **10f** by reducing the solvent exposure of amide protons and facilitating their transfer from the polar exterior to the nonpolar interior of the membrane with the adoption of a membranophilic conformation.

3) The NMR based conformational analysis discussed on page 8 is a) based on data that is prone to be influenced by experimental error and b) has been conducted using a method that drives the analysis towards one virtual (unreal) conformation, or a set of similar virtual conformations.

a) Interatomic distances cannot be accurately determined from a single NOESY or ROESY spectrum with an arbitrary mixing time. Instead NOESY buildups with several different mixing times should be acquired, then the linear part of the initial build-ups can be used to determine distances. Use of ROESY buildups is cumbersome as these need correction for offset effects and may suffer from TOCSY-type artifacts for strongly coupled proton pairs.

b) With very few exceptions NMR spectra acquired at room temperature constitute a time-average from multiple rapidly equilibrating conformations. Consequently, they need to be deconvoluted into the contributions from the individual conformations and not interpreted as a single, virtual (average) conformation that satisfies all NOE restraints (Cicero, et al., J. Org. Chem. 1999, 64, 11, 3979). Use of NMR derived restraints in MD simulations, as done in the manuscript, will provide such a virtual conformation. Deconvolution of the NMR data into real conformations can be done using Mestrenova's Stereofitter or the NAMFIS algorithm (Cicero, et. al., J. Am. Chem. Soc. 1995, 117, 1027).

Either the NMR-based conformational analysis should be redone using a state-of-the-art method or removed. An unrestrained MD simulation of each cyclic peptide might be just as informative.

Thank you for the valuable insights and the suggestion, which has tremendously improved this sub-section. We have redone the conformational analysis through unrestrained MD-simulation run over 200 ns. We do observe differences between the average conformation obtained from the NOE-derived restrained MD simulation and unrestrained MD simulation. Remarkably, the conformations derived from the unrestrained MD simulation beautifully justify the HDX values obtained from the NMR studies. Thus, we describe and compare the conformations obtained from the unrestrained MD simulation.

*We have accordingly incorporated the results in **Figs. 2D-K**.*

Fig.2. The average solution structure determined by 200 ns fMD simulation of (D) **10**, (E) **10a**, (F) **10e**, and (G) **10f** in DMSO- d_6 . The amino acid residue numbers are shown only in **10**. The backbone superimposed average structures obtained from the 200 ns rMD (iron) and fMD (aqua) simulation of (H) **10**, (I) **10a**, (J) **10e**, and (K) **10f**. The Ramachandran plot shows the deviation in torsion angles (ϕ, ψ) of the residues in the average structures obtained from the rMD and fMD simulation by the broken arrows.

The following text from pages 10-11 describes the results.

Structural impact of thioamidation. To understand the conformational impact of thioamide substitution, the solution structures of **10**, **10a**, **10e**, and **10f** in DMSO- d_6 was determined through restrained molecular dynamics (rMD) simulation by using the interproton distances derived from the ROESY spectrum (Fig. S6). An unrestrained MD (fMD) simulation was also performed to assess the similarity of the structures (Fig. 2D-2G) to the ones derived from experimental restraints. Thioamide substitution either at an externally oriented (**10a**, **10e**) or internally oriented C=O (**10f**) did not introduce a cis-peptide bond, unlike N-methylation³⁴. The RMSD calculated over $C\alpha$ atoms within **10** and **10a** (0.71 Å), **10e** (0.85 Å), **10f** (0.41 Å) suggests that there are no major overall structural change on thioamidation (Fig. S7 & S8). However, we note that

thioamide modulates the local conformation within the macrocyclic peptide through steric effects.

10 shows a structurally rigid type II' β turn about D-Pro^{4ⁱ⁺¹}-Tyr^{5ⁱ⁺²} (Fig. 2D) that is stabilized by Leu3CO...HNLeu6 H-bond, resulting in slow exchange of Leu6NH (Fig. 2B). On the contrary, the β -turn about D-Leu^{1ⁱ⁺¹}-Leu^{2ⁱ⁺²} is distorted due to the outward flip of Leu6CO, perhaps to pack the side chains of Leu6 and Leu2. This results in the lack of Leu6CO...HNLeu3 H-bond causing rapid exchange of Leu3NH (Fig. 2B). The thioamide substitution at D-Leu1 (**10a**) results in an upward flip of the C=S to avoid the steric clash between S and D-Leu1C ^{β} (Fig. 2E). Consequently, we note an alteration in the backbone torsion angles (ϕ, ψ) of D-Leu1 and Leu2 that are associated with the thioamide bond (Table S3)³⁵. The C=S orientation also alters the χ_1 of Leu2 to a *trans* conformation (as opposed to *gauche*⁺ in **10**) to avoid steric clash between S and the isopropyl group. This ultimately creates a hydrophobic patch formed by Leu2 and Leu3 side chains around Leu3NH (Fig. 2E), resulting in its slow exchange (Fig. 2B). The thioamide incorporation at Tyr5 in **10e**, forces the Leu6CO inwards to avoid the steric clash between C(S) and Leu6C(O) resulting in a type II' β -turn about D-Leu^{1ⁱ⁺¹}-Leu^{2ⁱ⁺²} (Fig. 2F & Table S3). However, the rapid exchange of Leu3NH suggests the absence of an intramolecular H-bond stabilising the β -turn, in contrast to the β -turn at D-Pro^{4ⁱ⁺¹}-Tyr^{5ⁱ⁺²} stabilized by Leu6CO...HNLeu3 H-bond (Fig. 2B). Additionally, the χ_1 of Tyr5 is in a *trans* conformation instead of *gauche*⁺ in **10**, perhaps to optimize the hydrophobic packing within the aromatic ring and S. Thioamidation at Leu6 (**10f**), results in a structure with maximal resemblance to **10** (Fig. 2G). The local desolvation at the C(S) and its hydrophobic interaction²⁷ with Leu2 side chain results in solvent shielding of Leu2NH (Fig. 2B). In addition, the unusually slow exchange of

Tyr5NH presumably results from the formation of an intramolecular H-bond with Leu3CO resulting in a γ -turn.

To further assess the influence of thioamidation on the conformational flexibility of **10**, the average structures resulting from the restrained and free MD simulations were compared. We note large alterations in the backbone torsion angles of **10** within the two structures, including the residues at the type II' β -turn (D-Pro4 and Tyr5) (Fig. 2H). However, **10a**, **10e**, and **10f** shows low variation in the torsion angles of Leu3, D-Pro4, and Tyr5 as compared to **10** (Fig. 2I-2K). This result suggests that thioamide substitution can induce segmental rigidity into macrocyclic peptides. The reduced conformational flexibility on thioamidation was also indicated by a reduction in the spread of the dihedral angles ϕ and ψ during the free MD simulation in both polar and nonpolar solvents (Fig. S9-S11). Therefore, the restricted conformational freedom resulting in enhanced solvent shielding of the amide protons contribute to the improved passive permeability of the macrocyclic peptide on thioamidation³⁶.

4) Figures S6-S10, S14-S16 and S19-S24 are missing in the Supporting Information.

We are extremely sorry for this, as the automated .pdf conversion during the submission of the manuscript led to this error. We have now taken extra care to upload the files during the revision.

5) The chromatograms of cyclic peptides 10c, 10d and 11f contain large impurities close to the main peaks (5-10%?). Presumably the impurities originate from closely related peptides. I am not too worried that this will have influenced the conclusions in the manuscript, but it will have to be an editorial decision whether or not to accept the impurities.

As the peptides were synthesized multiple times, we erroneously exported the HPLC chromatograms of the peptides with low purity. We have now corrected this.

Minor

1) Error bars are given in most figures. The number of repeats should be given in the legend to further inform the reader about the quality of the experimental data.

We have now included the number of repeats in each experiment.

2) Page 5, lines 126-128 and Figure 1A. I do not agree that HBA masking has a greater impact

on lipophilicity than HBD masking for 1–7. This is only the case for compound 2, whereas differences between HBA and HBD masking are not significant for 1, 3, 4 and 7. For 5 and 6 differences are barely significant, but go either way. My conclusion is that HBA and HBD masking in this dipeptide model have an equal effect on lipophilicity.

Thank you for highlighting the point. As the AlogP and the C18 retention times for most of the thioamidated analogs are higher than the respective N-methylated counterparts, we chose to conclude that HBA masking has a greater impact on lipophilicity than HDB masking. However, we agree that the differences are not significant in this model dipeptide and thus have corrected the statement.

“Irrespective of the polarity of dipeptides 1-7, we noted a significant increase in the C18 retention time and logD_{7.4} of 1a-7a and 1b-7b indicating their enhanced lipophilicity. This suggests that HBA and HBD masking in this model dipeptide have a comparable effect on lipophilicity.”

3) Page 6, line 142. The solubility of 10 and 10a-f is mentioned, but no reference is given to Figure S3A which shows the data.

This has been corrected, and the data is shown in Fig S3C.

4) Figure 1H. Caco2 cell permeability has only been determined for 10a, 10e and 10f which have high PAMPA permeabilities. I recommend that the Caco2 permeability is also determined for one of the three cyclic peptides that have low PAMPA permeability, e.g. 10d, to investigate the correlation between PAMPA and Caco2 permeabilities better.

Thank you for suggesting this; we have now determined the bidirectional Caco-2 permeability of the low permeable 10c, which verify the correlation between the PAMPA and Caco-2 permeabilities.

*Please refer to **Figs. 3C&D** above.*

5) The MD simulations illustrated in Figure 2D have only been conducted for 2 ns. This appears to be a very short time for a cyclic peptide for which rotation about amide bonds is restricted.

Thank you for suggesting this. The runs have now been performed for 200 ns, and the results correlate well with the experimental observables.

6) Page 9, lines 219-222. a) I assume that line 220 should read "hypothesis that reduced desolvation of the...". b) Similarly, I assume that line 221 should read "through increased solvent shielding of the..".

The sentence has been corrected:

"This supports the hypothesis that reduced desolvation of the amide bonds directly induced by thioamidation and indirectly through increased solvent shielding of the amide protons is the major driver of passive transcellular permeability."

7) Page 17, lines 429-431. One gets the impression that N-methylation of an amide bond introduces conformational restriction when reading this sentence, but the opposite is true, i.e. that N-methylation results in increased conformational flexibility as both rotamers about the amide bond become populated.

The sentence has been corrected:

"However, it introduces conformational restriction, a feature that is in stark contrast to N-methylation, which introduces conformational flexibility by lowering the cis/trans rotational barrier."

Reviewer #3 (Remarks to the Author):

This paper by Ghosh et al describes the substitution of oxygen by sulphur in peptide bonds, in order to improve the pharmacokinetic properties of (macrocyclic) peptides. The results are clearly described and the overall message (improved proteolytic stability and membrane passage) is generally well-founded. Even though the data are promising, the paper is still premature and additional experiments are needed to improve the manuscript.

We greatly appreciate the valued opinion of the reviewer to add additional data to demonstrate the strength of the approach for wider reach and use by colleagues and other researchers.

- More specifically, the genericity of the approach is not demonstrated and taken together, the title is not supported enough as only one particular type of cyclic peptide has been investigated, namely one type of cyclic hexapeptide (type 9/10/11 in manuscript).

We absolutely agree with the reviewer that the demonstration of our approach only in a cyclic hexapeptide scaffold raises concern over its utility in macrocyclic peptides with varying ring sizes and composition.

Thus, to demonstrate the utility of thioamidation in improving the permeability of macrocyclic peptides of varying size and shape, we selected two additional scaffolds:

*11: cyclo(-Ile¹-Ala-Ala-Phe-Pro-Ile-Pro⁷-)¹² (7-mer), that was shown to display low permeability (reported in Nielsen, D.S. et al. Improving on Nature: Making a Cyclic Heptapeptide Orally Bioavailable. *Angew. Chem. Int. Ed.* **53**, 12059-12063 (2014)), and*

*12: cyclo(-D-Leu¹-Leu-D-Pro-D-Leu-Leu-D-Ala-Pro-Leu⁸-)¹⁰ (8-mer), a computationally designed highly permeable cyclic peptide with maximized intramolecular H-bond satisfaction (reported in Bhardwaj, G. et al. Accurate de novo design of membrane-traversing macrocycles. *Cell* **185**, 3520-3532 (2022)).*

We found that thioamidation in either scaffold led to a marked improvement in its lipophilicity, and subsequently, the permeability of the thioamidated analogs was several folds higher than that of the parent scaffold. Our results demonstrate that

- *the absolute permeability of a thioamidated macrocycle scales with lipophilicity ($\log D_{7.4}$) of the parent macrocycle, which is dictated by its composition and not the ring size, and*
- *H-bond acceptor (HBA) masking through thioamidation is a powerful tool to further navigate the permeability space beyond the currently available toolkit.*

*The results of this additional study are included in **Fig. 1** and the accompanying text on **pages 5-7**.*

Fig.1. (A) Octanol-water partition coefficient ($\log D_{7.4}$) of oxo (grey), thioamidated (yellow), and N-methylated (blue) dipeptides with the common sequence FX. The "X" residues are mentioned at the bottom of the bars. (B) $\log D_{7.4}$ of the cyclic pentaalanine (**8**) and hexaalanine (**9**) peptides with their respective thioamidated analogs indicated by small letters, which denote the site of thioamidation. (C) AlogP of cyclo(-D-Leu¹-Leu-Leu-D-Pro-Tyr-Leu⁶-)(**10**), cyclo(-Ile¹-Ala-Ala-Phe-Pro-Ile-Pro⁷-)(**11**), and cyclo(-D-Leu¹-Leu-D-Pro-D-Leu-Leu-D-Ala-Pro-Leu⁸-)(**12**). The small letters denote the site of thioamidation in the individual scaffolds. (D) $\log D_{7.4}$ of **10**, **11**, **12** and their respective thioamidated analogs. (E) Membrane permeability of the macrocyclic peptides determined by the PAMPA (P_e). Propranolol and **PC** (cyclo(-D-Leu¹-Leu-D-Leu-Pro-Tyr-D-Leu⁶-))³¹ were used as the markers for transcellular transport. (F) The plot of $\log D_{7.4}$ vs. PAMPA permeability of all the 24 macrocyclic peptides. The blue shaded region highlights the lipophilicity zone of macrocycles with permeability $> 2.5 \times 10^{-6}$ cm/s. $n = 3 \pm \text{SEM}$. Statistical significance was measured against the (all-amide) parent molecule by a one-tailed unpaired t-test. * $P < 0.05$, ** $P < 0.01$, *** $P < 0.001$, **** $P < 0.0001$, ns (non-significant) > 0.05 .

Impact of thioamidation on passive permeability of macrocyclic peptides. Since lipophilicity shows a strong positive correlation with cellular permeability of non-peptidic macrocycles³⁰, we sought to assess if this single atom substitution could improve the passive transcellular permeability of macrocyclic peptides. To test this, we chose three macrocyclic peptides cyclo(-D-Leu¹-Leu-Leu-D-Pro-Tyr-Leu⁶-)³¹ (6-mer) (**10**), cyclo(-Ile¹-Ala-Ala-Phe-Pro-Ile-Pro⁷-)¹² (7-mer) (**11**), and cyclo(-D-Leu¹-Leu-D-Pro-D-Leu-Leu-D-Ala-Pro-Leu⁸-)¹⁰ (8-mer) (**12**) with varying size, composition,

and lipophilicity (AlogP) (Fig. 1C). We synthesized their monothionated regioisomers **10a-10f**, **11a-11g**, **12a-12h** (Table S1) and assessed the aqueous solubility, retention in the C18 column, and $\log D_{7.4}$ to evaluate the impact of thioamidation (Fig. S3B-3C & 1D). In all the three scaffolds, thioamidation led to reduced aqueous solubility and enhanced retention in the C18 column. Furthermore, these two parameters correlated well (Fig. S3D-3E) with the octanol-water distribution coefficient ($\log D_{7.4}$) indicating enhanced lipophilicity of the macrocyclic scaffolds on thioamidation, irrespective of their size and composition. Encouraged by this, the passive permeability of the analogs was evaluated using an artificial membrane (PAMPA), which measures permeability of compounds in the absence of transporters and efflux systems³². In all three scaffolds, thioamidation resulted in a significant increase in the permeability of monothioamidated analogs (Fig. 1E). Although the absolute permeabilities of **11a-11g** are lower than **10a-10f** and **12a-12h**, the relative change in permeability of **11a-11h** with respect to **11**, is comparable to the other two scaffolds (Table S2). This suggests that thioamidation can improve the passive permeability of macrocycles with varying lipophilicity. However, the absolute permeability of the thioamidated macrocycle scales with lipophilicity ($\log D_{7.4}$) of the parent macrocycle, which is dictated by its composition and not the ring size (Fig. 1F). Nevertheless, a large increase in lipophilicity (e.g. **12c**) can potentially limit the passive permeability of macrocyclic peptides, as observed before³³. It is also important to note that **12**, referred as D8.1 by Bhardwaj et al. displayed the highest permeability within the computationally designed 8-mer scaffolds due to the maximisation of intramolecular H-bond satisfaction¹⁰. However, the enhanced permeability of **12** through thioamidation (**12d-**

12h) strongly justifies the potential of HBA masking to further navigate the permeability space beyond the currently available toolkit.

In addition, we have also synthesized all the N-methylated analogs of 10 (10g-10k) and performed the detailed in vitro and in vivo pharmacokinetic (PK) studies for a pair-wise comparison of the effect of HBA masking by thioamidation vs. H-bond donor (HBD) masking by N-methylation. Our results suggest that single-site HBA masking is as potent as single-site HBD masking in improving the PK properties of macrocyclic peptides.

*The results of this study are compiled in **Fig. 4**, **Fig. 5** and **Table 1**. (Please see above for the figures and detailed discussion).*

- The diversity of tests presented is very relevant, but some techniques are not elaborated enough. For example, the conformational impact of the thioamide insertion is not clear. The conformational analysis is poorly described and figures in the ESI even lack. Hence, the conformational impact of the thioamide inclusion is not detailed enough to understand what this chemical modification does to the overall and/or local conformation. It is therefore strongly advised to improve these sections.

We apologize for the error in checking the .pdf file before the submission. It has been corrected now and has been checked thoroughly.

We strongly agree with the reviewer that a detailed conformational assessment would be valuable for the readers and will be helpful in guiding future designs. Thus, we have performed 200 ns restrained followed by 200 ns free MD simulation in polar and non-polar environments to derive the conformation of the cyclic peptides. It was gratifying to note that the results of the simulation agreed well with the experimental observables.

*The results are compiled in **Fig. 2** and the accompanying text on **pages 10-11**.*

Fig.2. (A) Chemical structure of **10**, with each amide proton color coded. The small letters denote the site of thioamidation. (B) Dot plots representing the half-lives ($t_{1/2}$) of the individual amide protons of **10**, **10a-f** deduced from the HDX experiment in 20% D_2O in $DMSO-d_6$, and (C) in 20% CD_3OD in $CDCl_3$. The slow exchanging amide protons whose $t_{1/2}$ could not be determined have been represented as % remaining after 24 hours. The average solution structure determined by 200 ns fMD simulation of (D) **10**, (E) **10a**, (F) **10e**, and (G) **10f** in $DMSO-d_6$. The amino acid residue numbers are shown only in **10**. The backbone superimposed average structures obtained from the 200 ns rMD (iron) and fMD (aqua) simulation of (H) **10**, (I) **10a**, (J) **10e**, and (K) **10f**. The Ramachandran plot shows the deviation in torsion angles (ϕ, ψ) of the residues in the average structures obtained from the rMD and fMD simulation by the broken arrows.

Structural impact of thioamidation. To understand the conformational impact of thioamide substitution, the solution structures of **10**, **10a**, **10e**, and **10f** in $DMSO-d_6$ was determined through restrained molecular dynamics (rMD) simulation by using the interproton distances derived from the ROESY spectrum (Fig. S6). An unrestrained MD (fMD) simulation was also performed to assess the similarity of the structures (Fig.

2D-2G) to the ones derived from experimental restraints. Thioamide substitution either at an externally oriented (**10a**, **10e**) or internally oriented C=O (**10f**) did not introduce a cis-peptide bond, unlike N-methylation³⁴. The RMSD calculated over C α atoms within **10** and **10a** (0.71 Å), **10e** (0.85 Å), **10f** (0.41 Å) suggests that there are no major overall structural change on thioamidation (Fig. S7 & S8). However, we note that thioamide modulates the local conformation within the macrocyclic peptide through steric effects.

10 shows a structurally rigid type II' β turn about D-Pro^{4 $i+1$} -Tyr^{5 $i+2$} (Fig. 2D) that is stabilized by Leu3CO \cdots HNLeu6 H-bond, resulting in slow exchange of Leu6NH (Fig. 2B). On the contrary, the β -turn about D-Leu^{1 $i+1$} -Leu^{2 $i+2$} is distorted due to the outward flip of Leu6CO, perhaps to pack the side chains of Leu6 and Leu2. This results in the lack of Leu6CO \cdots HNLeu3 H-bond causing rapid exchange of Leu3NH (Fig. 2B). The thioamide substitution at D-Leu1 (**10a**) results in an upward flip of the C=S to avoid the steric clash between S and D-Leu1C β (Fig. 2E). Consequently, we note an alteration in the backbone torsion angles (ϕ, ψ) of D-Leu1 and Leu2 that are associated with the thioamide bond (Table S3)³⁵. The C=S orientation also alters the χ_1 of Leu2 to a *trans* conformation (as opposed to *gauche*⁺ in **10**) to avoid steric clash between S and the isopropyl group. This ultimately creates a hydrophobic patch formed by Leu2 and Leu3 side chains around Leu3NH (Fig. 2E), resulting in its slow exchange (Fig. 2B). The thioamide incorporation at Tyr5 in **10e**, forces the Leu6CO inwards to avoid the steric clash between C(S) and Leu6C(O) resulting in a type II' β -turn about D-Leu^{1 $i+1$} -Leu^{2 $i+2$} (Fig. 2F & Table S3). However, the rapid exchange of Leu3NH suggests the absence of an intramolecular H-bond stabilising the β -turn, in contrast to the β -turn at D-Pro^{4 $i+1$} -Tyr^{5 $i+2$} stabilized by Leu6CO \cdots HNLeu3 H-bond (Fig. 2B). Additionally, the χ_1 of Tyr5 is in a *trans* conformation instead of *gauche*⁺ in **10**, perhaps

to optimize the hydrophobic packing within the aromatic ring and S. Thioamidation at Leu6 (**10f**), results in a structure with maximal resemblance to **10** (Fig. 2G). The local desolvation at the C(S) and its hydrophobic interaction²⁷ with Leu2 side chain results in solvent shielding of Leu2NH (Fig. 2B). In addition, the unusually slow exchange of Tyr5NH presumably results from the formation of an intramolecular H-bond with Leu3CO resulting in a γ -turn.

To further assess the influence of thioamidation on the conformational flexibility of **10**, the average structures resulting from the restrained and free MD simulations were compared. We note large alterations in the backbone torsion angles of **10** within the two structures, including the residues at the type II' β -turn (D-Pro4 and Tyr5) (Fig. 2H). However, **10a**, **10e**, and **10f** shows low variation in the torsion angles of Leu3, D-Pro4, and Tyr5 as compared to **10** (Fig. 2I-2K). This result suggests that thioamide substitution can induce segmental rigidity into macrocyclic peptides. The reduced conformational flexibility on thioamidation was also indicated by a reduction in the spread of the dihedral angles ϕ and ψ during the free MD simulation in both polar and nonpolar solvents (Fig. S9-S11). Therefore, the restricted conformational freedom resulting in enhanced solvent shielding of the amide protons contribute to the improved passive permeability of the macrocyclic peptide on thioamidation³⁶.

- Additionally, it is not described how this specific modification influences the proteolytic stability, since it is not mentioned 'where' the protection takes place, meaning which amide/peptide bonds are positively impacted by this modification. Is it only the thioamide bond itself, or the adjacent ones as well? No metabolites are provided, neither for the SGF, not for the SIF or plasma stability assays.

*This is an extremely valuable suggestion. However, even after numerous digestion trials of **10**, **10a-10f** (thioamidated), and **10g-10k** (N-methylated) with SGF & SIF, we failed to identify the primary metabolites due to multiple cleavage sites on the macrocyclic peptide resulting in linearized fragments that were instantaneously cleaved into shorter fragments of identical masses (due to the presence of four leucine residues in a stretch). The presence of multiple cleavage sites on **10** is justified by the half-lives of **10a-10k**, where none of them have $t_{1/2} > 24h$. This is in stark contrast to **13**, cyclo(-D-Trp¹-Lys-Thr-Phe-Pro-Phe⁶-) ($t_{1/2} = 1.8 h$), where*

we could identify the cleavage site in SIF: the peptide bond between -Lys²-Thr³- by identifying the metabolite. And subsequently demonstrate the increase in $t_{1/2}$ to > 72 h by thioamide modification at the P1(Lys²) and P2 (D-Trp¹) sites, while no protection was offered by thioamidation at the P1' (Thr³) site.

Nevertheless, since we have obtained the metabolic stability for all the thioamidated and N-methylated analogs of **10**, we are in a position to identify the direct cleavage sites (P1), the proteolysis at which are slowed down (or prevented) by these two chemical modifications. With this information, we have described how thioamidation provides protection against proteolysis in **10**.

Figs. 4D & 4E and the accompanying text on **pages 15-16 & 24** describe the finding supported by the ESI **Fig. S25**.

Fig.4. 1A-6A refer to the amide bonds in **10**. The yellow and the blue bars in the individual panels represent the thioamidated (**10a-10f**) and the N-methylated (**10g-10k**) counterparts, respectively, at a specific amide bond. Half-lives ($t_{1/2}$) of the peptides in (D) simulated gastric fluid (SGF), and (E) simulated intestinal fluid (SIF). (*adopted from Fig. 4 above*)

Macrocyclic peptides are endowed with proteolytic stability associated with their restricted conformational freedom; however, we were keen to evaluate their stability under the harsh proteolytic condition in the gastrointestinal tract⁴¹. Despite the presence of two D-amino acid residues (D-Leu¹ and D-Pro⁴) that are usually considered to confer proteolytic resistance to peptides, **10** underwent rapid degradation in Simulated Gastric Fluid (SGF) ($t_{1/2}$ 12 min), consisting of only pepsin at pH 1.2⁴² (Fig. 4D & S14). Chemical modification of an amide bond targeted by a protease increases its half-life^{43,44}. Thus, a significant increase in the half-life of the

peptides by both thioamidation and N-methylation at the amide bonds 1A and 5A indicate that 1A and 5A are the direct cleavage sites of pepsin on the macrocycle, with 5A (flanked by Tyr⁵ and Leu⁶) being the primary cleavage site. Curiously, the increased half-life on thioamidation but not N-methylation at 6A suggests that it is not the direct cleavage site of pepsin. However, thioamidation at 6A perhaps slows down the cleavage at the preceding amide bond 5A and the following amide bond 1A, as observed earlier for linear peptides treated against serine proteases⁴³.

We next assessed the stability of the macrocyclic peptides in Simulated Intestinal Fluid (SIF)⁴², comprising of pancreatin, which is a mixture of several proteases produced by the exocrine cells of the porcine pancreas. While the parent peptide **10** showed low proteolytic stability ($t_{1/2}$ 16 min), the maximum protection was achieved by thioamidation at D-Leu¹ (**10a**) ($t_{1/2}$ 147 min), Tyr⁵ (**10e**) ($t_{1/2}$ 241 min), and Leu⁶ (**10f**) ($t_{1/2}$ 126 min) (Fig. 4E & S15), as observed in SGF. In addition to 1A and 5A, either chemical modification at 2A protects the peptide from rapid degradation, highlighting the direct cleavage sites in SIF. In contrast, the increase in half-life only by thioamidation at 4A and 6A points at the indirect protection offered by thionating the preceding and following residues of 5A. Nonetheless, the protection offered by thioamidation outweighs the effect of N-methylation. The combined results of SGF and SIF highlight the benefit of thioamidating hydrophobic amino acids to protect macrocyclic peptides against proteolysis.

Figure S25. (A) Chemical structure of **13**, with the site of cleavage (P1) between Lys² and Thr³ highlighted. P1', P1, and P2 sites which were thioamidated are highlighted in olive (**13c**), teal (**13b**), and orange (**13a**) respectively. (B) Degradation kinetics of **13**, **13a**, **13b**, and **13c** in SIF over 72 hours. (C) MALDI profile of **13** after incubation in SIF for 1 hour, showed both cyclic and linear mass. (D) MS-MS of the linear peak revealed the Lys²-Thr³ amide bond to be the primary site of cleavage. MALDI profiles of **13a** (E), **13b** (F), and **13c** (G) after incubation in SIF for 1 hour clearly shows that thioamidation confers protection against the intestinal proteases in **13a** (P2) and **13b** (P1), but no such protection in **13c** (P1').

Our results also show that thioamidation provides metabolic stability to macrocyclic peptides against degradation by the harsh enzymes present in the GI tract. We note that thioamidation at the proteolytically labile peptide bond or at the preceding/following residues confer protection against proteolysis (Figure 4D & 4E).

This was further verified by performing the metabolite analysis of **13** post SIF treatment, where we identified -Lys²-Thr³- peptide bond as the primary cleavage site resulting in $t_{1/2}$ of 1.8 h (Fig. S25). Thioamidation at the cleavage (P1) site (Lys², **13b**) and the preceding (P2) site (D-Trp¹, **13a**) prevented the degradation of the macrocycle over 72h, while thioamidation at the P1' site (Thr³, **13c**) did not significantly increase the half-life of **13c** ($t_{1/2}$ 2.5 h) (Fig. S25).

- Some parts of the manuscript can also not be judged as parts of the ESI are missing. The application on other types of linear and cyclic peptides is advised in order to prove the more generic advantage structurally diverse peptides may have through this change in the peptide backbone.

We are extremely sorry for the difficulty the reviewer faced in assessing the manuscript due to missing figures in the ESI. This has been corrected. We also agree with the reviewer and thus have included data from additional experiments to prove the generic advantage of thioamidation on structurally diverse peptides (mentioned against the second query of the reviewer).

Minor comments are:

- sentence on lines 4-6 of page 6 needs revision, as it makes no sense

The sentence has been corrected to:

"Encouraged by this, the passive permeability of the analogs was evaluated using an artificial membrane (PAMPA), which measures the permeability of compounds in the absence of transporters and efflux systems³²."

- inclusion or reference to the synthetic protocols of the thioamidated sequences

We have included a brief section in the Methods (main text) and detailed section in the ESI on the synthesis of thioamidated sequences and appended a reference as well.

- page 5: cyclopenta- or cyclohexa- (not cyclicpenta...)

Corrected.

- page 5: last line when referring to (Fig 1C and 1D), S2 and S3A lack. The HPLC gradient should also be stated in the caption

*The data referred to in the sentence has been appended in **Figs. S3B** (C18 retention time), **S3C** (solubility), and **1D** ($\log D_{7.4}$). The HPLC gradients are also mentioned.*

Figure S3. (A) Chemical structures of peptides **11**, **10**, and **12**. HPLC retention time (B) in C18 column and % solubility (C) of peptides **10**, **11**, and **12** and their thio-analogs. (D) Linear correlation between $\log D_{7.4}$ and HPLC retention time of the peptides and their thio analogs. (E) Linear correlation between $\log D_{7.4}$ and % solubility of the same group of peptides.

- page 6: the B-A without Elacridar experiment is missing in the graph for the different analogues

The bidirectional permeabilities in the absence of Elacridar are added in Fig. 3C.

- page 7: not DMSO-d6 and CDCl3 were used, but mixes with D2O and MeOD-d4. This should be corrected in the text and the authors should explain why these mixes were used.

Thank you for pointing out the error. This has been corrected, and we have explained why the mixes were used.

- in figure 2D and in the main text, it should be mentioned which exact thioamide insertions conferred conformational stability

We have performed a detailed conformational analysis and have appended new data to explain the conformational stability in Figs. 2H-2K and Figs. S9-S11. The text also highlights the conclusion on the site-specific conformational rigidity.

- R2 value of 0.68 can hardly be named 'good'

This has been corrected with freshly done analysis through a newly performed experiment reported in Figs. 3A & 3B and the accompanying text on page 13.

Fig.3. (A) Partition coefficient of **10** and **10a-10f** in heptane-ethylene glycol (h/e) solvent system. (B) Linear correlation plot of $P_{e(\text{PAMPA})}$ vs. $\log D_{(h/e)}$.

Gratifyingly, the most permeable cyclic peptides **10a**, **10e**, and **10f** showed higher distribution coefficients indicating lower desolvation penalty than the rest (Fig. 3A). Consequently, an improved correlation was observed between PAMPA permeability and desolvation penalty (Fig. 3B) over lipophilicity (Fig. S12). This supports the hypothesis that reduced desolvation of the amide bonds directly induced by thioamidation and indirectly through increased solvent shielding of the amide protons is the major driver of passive transcellular permeability.

- several SXX figures lack in ESI

We apologize for this, and we have now taken extreme caution to prepare the ESI.

- in many graphs the statistical significance lacks (this needs serious attention)

We thank the reviewer for this suggestion, which helped us to make concrete suggestions based on statistical significance.

- page 10: the conclusion '...indicating the cleavage of every amide bond.' is not supported as no metabolic analysis is presented. The cleavage can occur at 1 or 2 spots only.

We thank the reviewer for the valuable insights. Our present data clearly indicate that cleavage occurs at the amide bonds 1A & 5A in simulated gastric fluid (SGF), and at 1A, 2A & 5A in simulated intestinal fluid (SIF) as shown in Figs. 4D & 4E and described in the accompanying text above.

- the error bars in Fig3D are too high, making the values and 'fold improvements' uncertain.

We have now determined the statistical significance of the area under the concentration-time curve (AUC) through a one-tailed unpaired t-test to justify the fold improvements noted in Table 1 (please see above).

- caption Fig4 should be revised. mg should be replaced by microgram, i guess. It is also stunning that SRIF has an effect of 24h, while it has a $t_{1/2}$ of 3 minutes. How is this possible?

Thank you for pointing out the error, which arose from the formatting of the text in the figure caption to Arial font. This has been corrected to microgram.

The $t_{1/2}$ of SRIF-14 in blood plasma is 3 minutes. However, we did not perform the in vivo PK study of SRIF-14 post s.c. injection to derive information about its release into plasma. The effect till 24 h is perhaps associated with its slow release post s.c. injection. In the absence of this data, we chose to substitute the GH inhibition release by SRIF-14 with an appropriate control 13a, which does not activate the SSTR2 and SSTR5 and shows prolonged plasma exposure.

Accordingly, we have modified Figs. 7E-7H and the accompanying text on page 23.

Fig.7. (A) Half-lives ($t_{1/2}$) of **13**, **13a-13f** determined in human blood plasma. (B) % protein binding of the peptides in plasma, as measured through RED. (C) The IC₅₀ values of the peptides determined from dose-dependent inhibition of forskolin-induced cAMP production in HEK293T cells expressing SSTR2 and SSTR5. $n = 3 \pm$ SEM. (D) Plasma concentration vs time plot of **13**, **13a**, **13b**, **13d**, and **13e** after a single dose (250 μ g/Kg) s.c.

administration in male Sprague-Dawley rats. The rapid drop in the concentration of **13** as opposed to its four thio analogs in plasma at later time points (2-8 h) are shown in the inset. (E) Inhibition of rat growth hormone (rGH) levels over 48 hours post single dose (250 μ g/Kg) peptide (**13**,**13a**,**13b**,**13d**,**13e**) and PBS (Untreated) administration via the s.c. route. (F) Bar plots represent the area under the concentration-time curve (AUCs) of rGH for (F) 0-48 h, (G) 0-24 h, and (H) 24-48 h. $n = 3 \pm$ SEM (A to D); $n = 6 \pm$ SEM (E). Statistical significance was measured against the (all-amide) parent molecule by a one-tailed unpaired t-test. * $P < 0.05$, ** $P < 0.01$, *** $P < 0.001$, **** $P < 0.0001$, ns (non-significant) > 0.05 .

In vivo inhibition of GH. Finally, we estimated the in vivo efficacy of **13** and the thioamidated analogs **13b**, **13d**, and **13e** in suppressing the release of growth hormone (GH) post subcutaneous injection at 250 μ g/kg dosage (Fig. 7E). **13a** was also included as a negative control to assess its effect on GH release. The rats were bled from their orbital sinus, and the plasma GH levels were evaluated through sandwich ELISA. Except **13a**, we observed a significant inhibition of unstimulated GH release by **13**, **13b**, **13d**, and **13e**. Remarkably, despite the strongest inhibition of GH release at the initial time point (1h) by **13**, the thioamidated analogs showed better inhibition of GH release at the subsequent time points throughout the course of experiment over 48 h (Fig. 7E). This correlates well with the high plasma concentration of **13** at the initial time points and a rapid drop within 2 h of s.c. administration (Fig. 7C). **13** has the least potency in inhibiting the GH release over 0-48 h (Fig. 7F), wherein it shows substantial inhibition only within 24h (Fig. 7G). In contrast, the inhibitory effect lasts longer in **13b**, **13d**, and **13e** showing a significant reduction in plasma GH levels within 24-48 h (Fig. 7H). **13d** shows the maximum inhibition of GH release, which is related to its longer retention in the plasma (Fig. 7C) due to its high plasma stability (Fig. 7A) than the other thioamidated analogs. The rapid drop in plasma concentration of **13** is responsible for its least in vivo efficacy. In comparison, the increased metabolic stability and lipophilicity of the macrocyclic peptide on thioamidation resulted in long-acting analogs that slowly diffused into plasma from the site of injection.

REVIEWERS' COMMENTS

Reviewer #1 (Remarks to the Author):

The authors did an excellent job on revising the manuscript and the new data has made the manuscript significantly better than the original submission. Importantly, the authors addressed the main concern by comparing their thioamidated peptides with its N-methylated counterparts. The authors have addressed all others concerns and, therefore, I recommend accepting this paper for publication in Nature Communications.

Reviewer #3 (Remarks to the Author):

The revised manuscript has been carefully composed and, as expected, the quality of the latter has significantly increased. Benchmarking the thioamide peptides with the 'golden standard' N-methylation was much appreciated and needed. However, the statement "Nonetheless, the protection offered by thioamidation outweigh the effect of N-methylation" should be revised to "... in these types of cyclic peptides". In fact, the authors cannot overstate the conclusions and this should be carefully addressed in the text, as the results might be linked to this particular type of macrocycle. When plotting the results, individual measurement points, and not only averages with standard deviations should be presented (eg Fig 5). Overall, the comments of this reviewer have been addressed and the broader applicability (other macrocycle sizes), as well as conformational impacts have been looked at. A point to address remains the reason why only PAMPA was used for the new macrocycles. Also, when hydrophobic patches arise in the modeling, an attempt to link these data to PPB might be undertaken.

REVIEWER COMMENTS AND POINT-WISE RESPONSE

Reviewer #1 (Remarks to the Author):

The authors did an excellent job on revising the manuscript and the new data has made the manuscript significantly better than the original submission. Importantly, the authors addressed the main concern by comparing their thioamidated peptides with its N-methylated counterparts. The authors have addressed all others concerns and, therefore, I recommend accepting this paper for publication in Nature Communications.

We sincerely wish to thank the reviewer for the careful read and the highly constructive comments that tremendously aided in demonstrating the uniqueness of Hydrogen Bond Acceptor masking by thioamidation in improving the permeability and bioavailability of macrocyclic peptides.

Reviewer #3 (Remarks to the Author):

The revised manuscript has been carefully composed and, as expected, the quality of the latter has significantly increased. Benchmarking the thioamide peptides with the 'golden standard' N-methylation was much appreciated and needed.

We greatly appreciate the reviewer's patience and efforts in helping us bring out the study's strength.

However, the statement "Nonetheless, the protection offered by thioamidation outweigh the effect of N-methylation" should be revised to "... in these types of cyclic peptides". In fact, the authors cannot overstate the conclusions and this should be carefully addressed in the text, as the results might be linked to this particular type of macrocycle.

We agree with the reviewer and have revised the statement. Also, we have checked the manuscript thoroughly to avoid overstating our conclusions.

When plotting the results, individual measurement points, and not only averages with standard deviations should be presented (eg Fig 5). Overall, the comments of this reviewer have been addressed and the broader applicability (other macrocycle sizes), as well as conformational impacts have been looked at. A point to address remains the reason why only PAMPA was used for the new macrocycles. Also, when hydrophobic patches arise in the modeling, an attempt to link these data to PPB might be undertaken.

We have now presented the individual measurements in every figure, wherever applicable, along with the mean and standard deviations for the benefit of the readers.

Only PAMPA was used for the new macrocycles as this assay provides a precise estimate of the passive transcellular permeability of the macrocycles. In our experiments with the hydrophilic macrocycle 13 and its thioamidated analogs 13a-13f, we could obtain the permeabilities from PAMPA (see Fig. 6c) to evaluate the impact of thioamidation on the

passive transcellular permeability of the macrocycle. In contrast, the permeability of **13** and its thioamidated analogs could not be determined from the Caco-2 assay (Supplementary Table 6).

We agree with the reviewer that hydrophobic patches on the surface of macrocyclic peptides would influence plasma protein binding (PPB). Thus, we calculated the polar surface area of the structures **10**, **10a**, **10e**, and **10f** along with **13**, **13a-13f**. Although we see an apparent reduction in the polar surface area within the macrocycles on thioamidation (Fig. 1a,b), resulting from the origin of hydrophobic patches on the surface. Unfortunately a good correlation with the plasma protein binding could not be obtained (Fig. 1c,e). This was a thought-provoking question, and we will address this in our future work with a more extensive library size of structurally analogous compounds and detailed hydrophobic patch calculations.

Fig 1. The polar surface area (PSA) of the a) hydrophobic and b) hydrophilic macrocycles calculated with QikProp module of Schrödinger suite on the 3D NMR structures. The correlation plot of plasma protein binding vs polar surface area of the c) hydrophobic and d) hydrophilic macrocycles.